# Shaving Weights with Occam's Razor: Bayesian Sparsification for Neural Networks using the Marginal Likelihood

**Rayen Dhahri**[1,2], **Alexander Immer**[3,4]**, Betrand Charpentier**[5]**, Stephan Günnemann**[1,2]**, and Vincent Fortuin**[1,2,6]

[1]Department of Computer Science, Technical University of Munich
[2]Munich Center for Machine Learning
[3]Department of Computer Science, ETH Zürich
[4]Max Planck Institute for Intelligent Systems, Tübingen
[5]Pruna AI, Munich
[6]Helmholtz AI, Munich

## Abstract

Neural network sparsification is a promising avenue to save computational time and memory costs, especially in an age where many successful AI models are becoming too large to naïvely deploy on consumer hardware. While much work has focused on different weight pruning criteria, the overall *sparsifiability* of the network, i.e., its capacity to be pruned without quality loss, has often been overlooked. We present ***Spa**rsifiability via the **M**arginal likelihood (SpaM)*, a pruning framework that highlights the effectiveness of using the Bayesian marginal likelihood in conjunction with sparsity-inducing priors for making neural networks more sparsifiable. Our approach implements an *automatic Occam's razor* that selects the most sparsifiable model that still explains the data well, both for structured and unstructured sparsification. In addition, we demonstrate that the pre-computed posterior precision from the Laplace approximation can be re-used to define a cheap pruning criterion, which outperforms many existing (more expensive) approaches. We demonstrate the effectiveness of our framework, especially at high sparsity levels, across a range of different neural network architectures and datasets.

## 1 Introduction

The availability of large datasets and powerful computing infrastructure has fueled the growth of deep learning, enabling the training of increasingly complex neural networks (NNs). While catalyzing performance gains across various domains, such as image recognition [1] and text generation [2], this development has amplified the challenge of over-parameterization [3, 4] and raised concerns about the increase in model size and computational cost. Over-parameterized neural networks present significant deployment challenges, particularly in hardware-constrained environments [5, 6]. This has sparked the research field of NN *sparsification* or *pruning*, where the goal is to remove a (potentially large) number of parameters from a trained network to make it smaller and ultimately cheaper to apply [7, 8]. However, most existing research in this domain has focused on the question of finding better pruning criteria, that is, scoring functions that decide which parameters to prune away [9, 4, 10]. This ignores the challenge that many trained networks are not inherently *sparsifiable*, i.e., they resist

---

Correspondence to rayen.dhahri@tum.de

38th Conference on Neural Information Processing Systems (NeurIPS 2024).

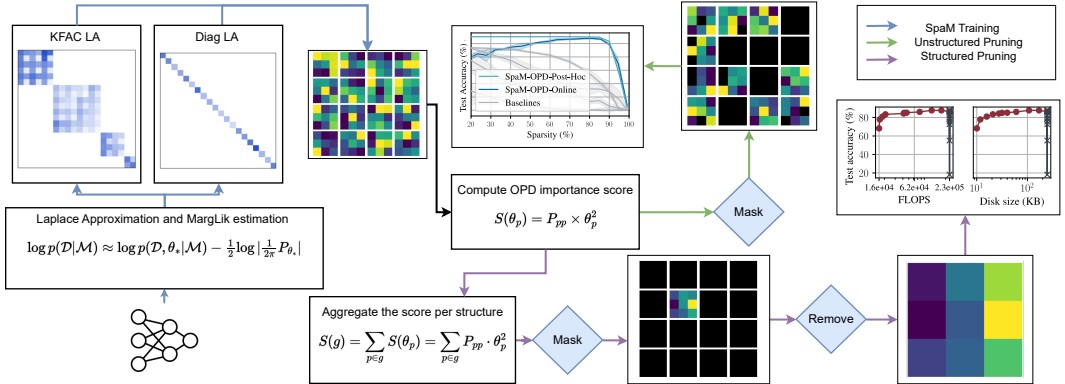

Figure 1: Overview of our proposed SpaM method. We start by training the network to maximize the marginal likelihood using the Laplace approximation, while simplifying the Hessian computation through either the KFAC or a diagonal approximation. We can then use our precomputed posterior precision as a pruning criterion (OPD). For the case of unstructured pruning, we compute thresholds to achieve different target sparsities, compute the mask, and apply it, while for the structured approach, we aggregate the score per layer for easier weight transfer, compute the mask, and then delete the masked structures to obtain a smaller model.

effective pruning, regardless of the chosen criterion. Indeed, standard training of NNs does not encourage sparsifiability at all, so it should not be surprising that such trained NNs would use all of their parameters to some degree to fit the data.

Our work tackles this problem by *modifying the training process itself*, showing that more *sparsifiable* networks can be achieved through Bayesian model selection using the marginal likelihood [11, 12] in conjunction with an adequate prior that will induce such sparsifiability. We call this *Sparsifiability via the Marginal likelihood*, or *SpaM*. Our approach implements an *automatic Occam's razor* [13], guiding the training process towards models that are faithfully fitting the data using only a small subset of their available parameters, such that the remaining ones can be pruned afterwards. This is achieved by optimizing thousands of prior hyper-parameters to adaptively regularize weight magnitudes. We make use of recent advances in Laplace inference for Bayesian neural networks (BNNs) [12, 14], allowing us to approximate the marginal likelihood [11] efficiently.

Once trained, we can use any pruning criterion to more effectively sparsify these networks. Notably, the pre-computed posterior precision of the Laplace approximation obtained from the marginal likelihood training readily translates into a powerful pruning criterion, which we call *Optimal Posterior Damage* (OPD), similar to the popular Optimal Brain Damage [OBD; 3]. Since it reuses existing computations, it is cheaper than many existing criteria in practice, and it often performs on par or even better.

Extensive empirical evaluations demonstrate the strength of our SpaM approach and the derived OPD pruning criterion in both unstructured and structured sparsification tasks across various datasets and architectures. Moreover, they show that our framework strikes a compelling balance between performance and computational cost.

We make the following contributions:

- We propose **SpaM**, a *novel approach to improve the sparsifiability of neural networks during training*, using Bayesian model selection with the Laplace-approximated marginal likelihood, which works well in *structured and unstructured pruning* scenarios and with *different pruning criteria*.

- We provide evidence-based *recommendations for prior selection* within the SpaM framework, showing that some priors can improve sparsifiability significantly better than others.

- We present **OPD**, a *cheap pruning criterion* similar to the popular OBD, which performs comparably or better than many other criteria in practice and *conveniently reuses computations* performed in the context of SpaM.

## 2 Background

We use deep neural networks to model learning tasks with inputs $\mathbf{x}_n \in \mathbb{R}^D$ and targets $\mathbf{y}_n \in \mathbb{R}^C$ collected in a dataset $\mathcal{D} = \{(\mathbf{x}_n, \mathbf{y}_n)\}_{n=1}^N$ of $N$ pairs. A model is parameterized by weights $\boldsymbol{\theta} \in \mathbb{R}^P$, and maps from inputs to targets using the neural network function $\mathbf{f}_{\boldsymbol{\theta}}(\mathbf{x})$. Assuming the data are *i.i.d.*, we have a likelihood $p(\mathcal{D}|\boldsymbol{\theta}) = \prod_{n=1}^N p(\mathbf{y}_n|\mathbf{f}_{\boldsymbol{\theta}}(\mathbf{x}_n))$. We minimize the negative log likelihood, which corresponds to common losses like the cross-entropy in classification. Additionally, regularization in the form of weight decay is commonly used and corresponds to a Gaussian prior on parameters $p(\boldsymbol{\theta}) = \mathcal{N}(\boldsymbol{\theta}; \mathbf{0}, \mathrm{diag}(\boldsymbol{\delta}))$ with diagonal precision. We employ Gaussian priors due to their analytical tractability and seamless integration with the Laplace approximation, which requires differentiability. This is essential for maintaining computational efficiency in our framework to approximate the marginal likelihood in a practical and scalable manner. Furthermore, Gaussian priors enable automatic relevance determination by allowing each parameter to have its own variance, facilitating the regularization process without introducing significant computational overhead.

### 2.1 Marginal Likelihood for Deep Learning

**The marginal likelihood** serves as the probabilistic foundation for model evaluation and selection. It provides an objective to optimize the tradeoff between data fit and model complexity, akin to the concept of Occam's razor [13, 15], by quantifying how well a model $\mathcal{M}$, with all its inherent uncertainties, explains the observed data:

$$p(\mathcal{D}|\mathcal{M}) = \int p(\mathcal{D}|\boldsymbol{\theta}, \mathcal{M})\, p(\boldsymbol{\theta}|\mathcal{M})\, \mathrm{d}\boldsymbol{\theta}. \tag{1}$$

However, it requires computing an intractable integral over all neural network parameters.

**The Laplace approximation** [LA, 16] provides a tractable and effective approximation to the marginal likelihood for deep learning [12]. It arises from a second-order Taylor approximation around an estimate of the mode, $\boldsymbol{\theta}_*$, resulting in

$$\log p(\mathcal{D}|\mathcal{M}) \approx \log p(\mathcal{D}, \boldsymbol{\theta}_*|\mathcal{M}) - \tfrac{1}{2}\log|\tfrac{1}{2\pi}\mathbf{P}_{\boldsymbol{\theta}_*}(\mathcal{M})|, \tag{2}$$

where $\mathbf{P}_{\boldsymbol{\theta}_*}$ is the posterior precision given by the Hessian of the negative log joint distribution, $-\nabla_{\boldsymbol{\theta}}^2 \log p(\mathcal{D}, \boldsymbol{\theta}|\mathcal{M})$, evaluated at $\boldsymbol{\theta}_*$. Defining $\mathbf{H}_{\boldsymbol{\theta}_*}$ as the Hessian of the negative log likelihood objective $-\nabla_{\boldsymbol{\theta}}^2 \log p(\mathcal{D}|\boldsymbol{\theta}, \mathcal{M})$, the posterior precision decomposes as $\mathbf{P}_{\boldsymbol{\theta}_*} = \mathbf{H}_{\boldsymbol{\theta}_*} + \mathrm{diag}(\boldsymbol{\delta})$.

In practice, the Hessian of the negative log likelihood is often approximated by the positive semidefinite **generalized Gauss-Newton** [GGN, 17],

$$\mathbf{H}_{\boldsymbol{\theta}} \approx \sum_{n=1}^N \nabla_{\boldsymbol{\theta}} \mathbf{f}_{\boldsymbol{\theta}}(\mathbf{x}_n) \nabla_{\mathbf{f}}^2 \log p(\mathbf{y}_n|\mathbf{f}_{\boldsymbol{\theta}}(\mathbf{x}_n)) \nabla_{\boldsymbol{\theta}}^\mathsf{T} \mathbf{f}_{\boldsymbol{\theta}}(\mathbf{x}_n), \tag{3}$$

which relies on the Jacobians of the neural network function and second derivative of the negative log likelihood at the output. Further, it is amenable to efficient structured approximations like diagonal or layer-wise variants [e.g., 18, 19].

**Diagonal and block-diagonal GGN approximations** are efficient, and therefore commonly used for Laplace approximations in deep learning [20, 14]. The diagonal LA is cheap in terms of storage and computation by only modeling the marginal variances of parameters. Kronecker-factored LA [KFAC LA, 20] instead relies on a block-diagonal approximation to the GGN of the parameters $\boldsymbol{\theta}_l$ in the $l$th layer,

$$\mathbf{H}_{\boldsymbol{\theta}_l} \approx \mathbf{A}_l \otimes \mathbf{G}_l, \tag{4}$$

where the factors are given by the outer products of pre-activations and Jacobians w.r.t. the output of a layer, respectively [18, 19]. Here, $A_l$ and $G_l$ are the uncentered covariances of the respective layer inputs and output gradients. The top left of Figure 1 shows a comparison of both structures.

### 2.2 Neural Network Pruning

The goal of the pruning procedure is to remove parameters from $\boldsymbol{\theta}$ without affecting the quality of the model output $\mathbf{f}_{\boldsymbol{\theta}}(\mathbf{x})$. While unstructured pruning consists in zeroing individual entries $\theta_p$ of the

weight matrices, structured pruning consists in deleting entire structured sets of parameters $g$, like rows or columns [21, 22]. The results of structured pruning enable smaller matrix multiplications that directly provide real-world efficiency gains on most hardware, including GPUs.

Pruning procedures usually follow three steps: **(1)** We use a scoring function $S(\cdot)$ to evaluate the importance of each individual parameter $S(\theta_p)$ for unstructured pruning, or of a structured set of parameters $S(g)$ for structured pruning. **(2)** We compute a binary mask $\mathbf{m}$ with the same dimensions as $\boldsymbol{\theta}$, which assigns 0 values to parameters whose unstructured or structured pruning scores are below a threshold $T$, and 1 otherwise. While the threshold $T$ is determined based on the target sparsity across layers for *global* pruning, it is determined per layer for *uniform* pruning [21]. **(3)** We apply the mask on the weight matrix with element-wise multiplication, $\mathbf{m} \circ \boldsymbol{\theta}$, to effectively remove the least important parameters. Alternatively, structured pruning enables us to directly remove rows or columns whose mask values are 0 to reduce weight matrix dimensions.

# 3   Shaving Weights with Occam's Razor

We identify sparsifiable neural networks by automatically regularizing (groups of) parameters to have small magnitudes, to facilitate pruning the least important ones, within a probabilistic framework. Specifically, we utilize priors that regularize parameters in potentially structured ways, leading to smaller magnitudes. To optimize the resulting prior hyperparameters, we employ the Bayesian marginal likelihood as a differentiable objective function, effectively implementing a Bayesian variant of Occam's razor that drives irrelevant parameters towards smaller absolute magnitudes. The regularized networks can then be pruned *with any method*. However, we additionally propose to reuse the computed posterior precision for sparsification as a cheap and effective criterion.

## 3.1   Structured Priors for Regularization

To reduce the magnitude of parameters and make them more amenable to pruning, we introduce structured priors and show how to combine them with diagonal and KFAC Laplace approximations. While a scalar prior, corresponding to weight decay, is the most common, it suggests that all parameters in a neural network are equally relevant and favors a uniform magnitude of parameters, which is suboptimal for pruning [23, Sec. 3.6].

Instead of scalar priors, we regularize parameters with different strengths using layer-, unit-, and parameter-wise priors. Layer-wise priors regularize individual layers differently and have been shown to aid pruning and improve generalization [12, 24, 25, 26]. Unit-wise regularization has been used mostly in traditional statistics, for example, for group sparsity [27], but recently also for channels or feature dimensions in neural networks [28, 29].

We consider different priors in the context of the Laplace approximation for marginal likelihood optimization and pruning: Scalar priors correspond to standard weight decay and are identical for all weights. Layer-wise priors provide a scalar regularizer $\delta_l$ per layer that is stacked into a vector $\boldsymbol{\delta}$ in line with the number of parameters per layer. Parameter-wise priors allow to specify $\boldsymbol{\delta}_p$ for each parameter $p$ individually. We define unit-wise priors so that each unit, which denotes a channel for convolutional and a hidden neuron for fully-connected layers, has a regularization strength for incoming and outgoing weights separately. Thus, a weight $\theta_p$ that connects unit $i$ at layer $l$-1 with unit $j$ in layer $l$ has prior $\mathcal{N}(0, [\delta_{l\text{-}1}]_i \cdot [\delta_l]_j)$, that is, each layer $l$ with $M_l$ hidden units has a prior vector $\boldsymbol{\delta}_l \in \mathbb{R}^{M_l}$. A weight is thus regularized more strongly whenever both its in- and output neurons are.

Our different priors are simple to combine additively with a diagonal Hessian approximation for the Laplace approximation (Equation (2)) but not with a KFAC structure. For that reason, so far, only scalar or layer-wise priors have been used for KFAC posterior approximations [14]. The main issue is that we have to preserve the Kronecker factors to keep the resulting memory cost low. For scalar or layer-wise priors, this can be achieved by an eigendecomposition of the individual factors

$$\mathbf{A} \otimes \mathbf{G} + \mathbf{I}\delta \overset{\text{def}}{=} \mathbf{Q}_A \boldsymbol{\Lambda}_A \mathbf{Q}_A^\mathsf{T} \otimes \mathbf{Q}_G \boldsymbol{\Lambda}_G \mathbf{Q}_G^\mathsf{T} + \mathbf{I}\delta = (\mathbf{Q}_A \otimes \mathbf{Q}_G)(\boldsymbol{\Lambda}_A \otimes \boldsymbol{\Lambda}_G + \mathbf{I}\delta)(\mathbf{Q}_A^\mathsf{T} \otimes \mathbf{Q}_G^\mathsf{T}), \ (5)$$

which means that the precision only needs to be added to the diagonal eigenvalues and no Kronecker product needs to be calculated for inversion or determinant calculation.

To add a diagonal prior precision $\boldsymbol{\delta}_l$ to the KFAC of the $l$th layer, we derive an optimal approximation in the KFAC eigenbasis, so as to maintain the Kronecker-factored structure of the posterior:

**Proposition 3.1** (Diagonal Prior in KFAC Eigenbasis). *Considering the Frobenius norm, the optimal diagonal perturbation of the KFAC eigenvalues $\mathbf{\Lambda}_A \otimes \mathbf{\Lambda}_B$ to add a diagonal prior precision is given by $\mathbf{\Lambda}_A \otimes \mathbf{\Lambda}_B + \hat{\boldsymbol{\delta}}$ with $\mathrm{mat}(\hat{\boldsymbol{\delta}}) = (\mathbf{Q}_G^T)^2 \mathrm{mat}(\boldsymbol{\delta})\mathbf{Q}_A^2$ where the square is element-wise and $\mathrm{mat}(\cdot)$ reshapes the vector to match the parameter shape used in KFAC. Thus, it can be computed efficiently without computing a Kronecker product.*

We provide the proof in Appendix A. The approach is similar to that of George et al. [30], who correct KFAC's eigenvalues towards the diagonal Gauss-Newton, but solves the problem of adding a full-rank diagonal instead of a rank-1 outer product to the KFAC eigenbasis.

## 3.2 Learning Regularization with the Marginal Likelihood

To optimize the potentially millions of regularization parameters, for example, arising from a parameter-wise prior, we employ the marginal likelihood as a differentiable objective. Optimizing regularization parameters has the advantage that different (groups of) parameters will be regularized differently and, therefore, become easier to prune. While it would be intractable to optimize that many regularization parameters using validation-based forms of optimization, the marginal likelihood can be estimated and differentiated during training [12, 31, 32].

Automatically determining the relevance of parameter-groups (ARD) is a common approach in Bayesian learning that can lead to sparsity and smaller parameter magnitudes [33, 34] and has been used especially in linear models. The marginal likelihood provides an objective that automatically regularizes irrelevant parameter groups more to lower their magnitude. Therefore, it implements a Bayesian variant of Occam's razor, finding the simplest model that explains the data well [13].

Mathematically, all the prior parameters $\boldsymbol{\delta}$ constitute the hyperparameters of the model $\mathcal{M}$ in the log marginal likelihood (Equation (1)) that we optimize interleaved with the neural network parameters. When optimizing the prior parameters, we use gradient ascent

$$\boldsymbol{\delta}_{t+1} \leftarrow \boldsymbol{\delta}_t + \alpha \nabla_{\boldsymbol{\delta}} \log p(\mathcal{D}|\boldsymbol{\delta})|_{\boldsymbol{\delta}=\boldsymbol{\delta}_t}, \tag{6}$$

or adaptive optimizers like Adam [35]. We follow Immer et al. [12] and optimize the Laplace approximation to the marginal likelihood after an initial burn-in phase with a certain frequency. We describe the optimization process and the related hyperparameters in Appendix D.4

## 3.3 Optimal Posterior Damage (OPD)

While sparsity regularization learned by marginal likelihood training can be advantageously combined with *any pruning criterion*, like Single-shot Network Pruning [SNIP; 36], variants of Gradient Signal Preservation [GraSP; 37, 38, 39], or magnitude pruning [7], we further propose a new pruning criterion that uses our Laplace approximation and extends the unstructured Optimal Brain Damage (OBD) pruning criterion [3]. While OBD traditionally uses the Hessian (approximation) $\mathbf{H}_{\boldsymbol{\theta}}$ of the loss, we propose to adapt it to use the posterior precision $\mathbf{P}_{\boldsymbol{\theta}}$, which additionally includes the prior precision $\boldsymbol{\delta}$. The importance score $S(\theta_p)$ for parameter $\theta_p$ becomes

$$S(\theta_p) = \mathbf{P}_{pp} \times \theta_p^2 \tag{7}$$

where $\mathbf{P}_{pp}$ denotes the posterior precision for the parameter $\theta_p$, extracted from the diagonal of the posterior precision matrix $\mathbf{P}_{\boldsymbol{\theta}}$. We call this novel posterior-based pruning criterion *Optimal Posterior Damage* (OPD). Intuitively, individual weights with high scores indicate certainty of the posterior distribution and a significant contribution to the model's functionality, as indicated by the magnitude.

We also propose a structured version of OPD by aggregating the score over a set of parameters $g$, i.e.,

$$S(g) = \sum_{p \in g} S(\theta_p) = \sum_{p \in g} \mathbf{P}_{pp} \times \theta_p^2 \tag{8}$$

In practice, the structured set of parameters $g$ corresponds to all parameters along one dimension of the weight matrix inside a layer, in order to reduce the size of the matrix multiplications. Since subsequent layers might have significantly different weight matrix dimensions impacting the magnitude of the aggregated sum, we opt for uniform structured pruning to guarantee a fair pruning treatment across all layers. This means we prune each layer by the same target percentage, reducing the dimensions of each layer by the same proportion relative to the unpruned model. This approach contrasts with

Table 1: Accuracies of pruned ResNets on CIFAR-10. The best training method for each pruning criterion is highlighted in green, where we see that SpaM improves performance for all criteria except the random baseline. The best performances overall at each sparsity level are shown in **bold**, showing that our cheap OPD criterion outperforms the others at high sparsities.

| Criterion | Training | Sparsity (%) | | | | |
|---|---|---|---|---|---|---|
| | | 80 | 85 | 90 | 95 | 99 |
| OPD | MAP | 88.06 (±0.12) | 82.32 (±0.44) | 64.08 (±1.32) | 37.52 (±2.34) | 17.32 (±1.01) |
| | SpaM | 90.78 (±0.66) | 90.78 (±0.65) | **90.68 (±0.65)** | **89.98 (±0.61)** | **66.28 (±5.89)** |
| GraSP | MAP | 82.87 (±0.48) | 68.78 (±1.88) | 48.65 (±2.69) | 26.46 (±1.86) | 15.75 (±0.80) |
| | SpaM | 91.50 (±0.66) | **90.94 (±0.65)** | 89.42 (±0.71) | 82.18 (±2.65) | 41.48 (±7.95) |
| SNIP | MAP | 53.96 (±2.72) | 37.74 (±2.21) | 26.74 (±3.17) | 13.88 (±0.87) | 12.58 (±0.36) |
| | SpaM | 67.40 (±5.68) | 52.62 (±6.84) | 33.75 (±5.71) | 17.06 (±2.23) | 11.90 (±0.51) |
| Magnitude | MAP | 88.17 (±0.12) | 81.92 (±0.37) | 61.60 (±1.11) | 32.88 (±1.52) | 16.12 (±0.90) |
| | SpaM | **91.55 (±0.64)** | 90.92 (±0.64) | 89.23 (±0.62) | 81.80 (±2.22) | 41.78 (±7.20) |
| Random | MAP | 11.25 (±0.48) | 12.15 (±0.92) | 11.65 (±0.62) | 10.45 (±0.17) | 10.27 (±0.17) |
| | SpaM | 11.00 (±0.48) | 10.47 (±0.86) | 10.56 (±1.15) | 10.01 (±0.45) | 9.81 (±0.61) |

achieving a global target sparsity that varies across layers, which can make it more difficult to consistently compress and adjust the model's size.

Moreover, as removing a full structure is more aggressive, we also apply gradual pruning during training. Finally, we omit pruning the final layer to mitigate overly strong impact on classification accuracy and computational stability [40].

When using our SpaM approach, the precomputed precision matrix from the Laplace approximation can be reused to compute OPD without computational overhead, in contrast to the other pruning criteria, which often require additional computations to be performed. Note that we will also show in our experiments that OPD additionally avoids the need for potentially expensive fine-tuning after pruning. Moreover, even in the case of maximum a posteriori (MAP) training, Laplace approximations of the inverse Hessian at $\theta_*$ can be additionally computed to approximate OPD. Finally, the OPD criterion can not only be computed *post-hoc* after training, but even *online* during training.

## 4 Related work

**Laplace-approximated BNNs.** From the early inception of Bayesian neural networks [41, 42], the Laplace approximation was a popular inference method [16]. In recent years, it has undergone a renaissance [18, 19, 20, 14], including critical work on using more scalable approximations for the associated marginal likelihood in the context of model selection [11, 12, 43], which we use in our framework. To the best of our knowledge, we are the first to study the benefits of this Laplace-approximated marginal likelihood in the context of sparsification of deep neural networks. However, similar methods that automatically quantify the relevance (ARD) of parameters have been derived and used for linear, bilinear, and kernel models [34, 44, 45] as an alternative to the Lasso. More recently, van der Ouderaa et al. [46] and Bouchiat et al. [47] used the ARD mechanism in deep learning to select layers and features by regularization, respectively.

**Pruning neural networks.** Various pruning criteria have been proposed to determine the importance of model parameters. Many criteria prune based on the weight magnitude [7, 48, 49] but usually required additional fine-tuning to recover accuracy. Sun et al. [8] proposed to combine activation and weight norms for pruning without fine-tuning. Other approaches include pruning using first-order information based on connectivity [36] or synaptic flow conservation [50], or second-order information aiming at preserving gradient flow [37, 38, 39]. Recently, van der Ouderaa et al. [51] focused on pruning LLMs based on a second-order Taylor expansion. In contrast, OPD uses second-order information provided by the posterior precision given by the Laplace approximation. Beyond pruning criteria, there have been many approaches to prune at initialization [36, 37, 50], during training [52, 53], and after training [7, 8]. In particular, multiple works proposed to leverage specific

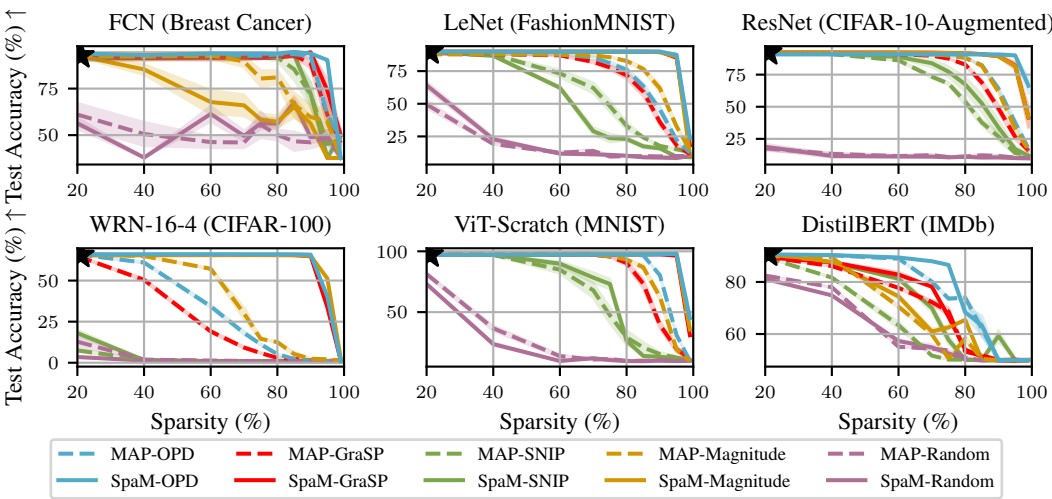

Figure 2: Predictive performance as a function of sparsity level in unstructured pruning. We see that SpaM improves the performance over MAP training across most architectures, datasets, and pruning criteria, and that OPD often outperforms the other pruning criteria. Both of these effects are particularly visible at higher sparsity levels. The black star in each subfigure denotes the performance of the unpruned models, which is often identical to the performance of models pruned at 20% sparsity.

training schemes promoting zero-invariant parameter groups for structured pruning [54, 55]. In contrast, SpaM induces sparsifiability during training, and is agnostic about the criterion.

## 5 Experiments

We conduct experiments on various datasets and models and outline our experimental setup in detail in Appendix D. We compare MAP training with our proposed SpaM approach with different priors, comparing our OPD pruning criterion with random pruning, magnitude pruning, SNIP [36], GraSP [37, 38], and SynFlow [50]. We show that **SpaM improves pruning performance with different pruning criteria**, especially at higher sparsities, and that **our OPD criterion often outperforms the other criteria**. This observation extends not only to predictive accuracy, but also uncertainty estimation. Moreover, we show that the choice of prior can play a significant role, and we **introduce parameter-wise and unit-wise priors** for the KFAC approximation. Finally, we show that SpaM and OPD also work in a structured pruning setting, leading to **significant computational benefits**. The code for our methods and experiments can be found at https://github.com/fortuinlab/spam-pruning.

### 5.1 SpaM Improves Performance at High Sparsities

We compare SpaM to MAP training with different pruning criteria, including OPD, across different models on tabular, vision, and language datasets. For SpaM in this unstructured pruning context, we use the diagonal Laplace approximation with a parameter-wise prior. Encouragingly, MAP and SpaM reach comparable performance during training, showing that the increased sparsifiability of SpaM comes at no additional cost in unpruned performance (see Figure B1 in the appendix).

We see in Table 1 and Figure 2 that SpaM drastically improves the performance for many pruning criteria, especially magnitude pruning, GraSP, and OPD. We also see that OPD, despite being a cheap byproduct of our marginal likelihood computation, often outperforms the other pruning criteria, especially at higher sparsities. For instance, at 95 % pruning rate (i.e., with 20x fewer parameters), our combination of SpaM and OPD still retains almost the same performance as the unpruned model on vision tasks, while the other pruning criteria with MAP training have dropped to essentially unusable performance levels at this sparsity.

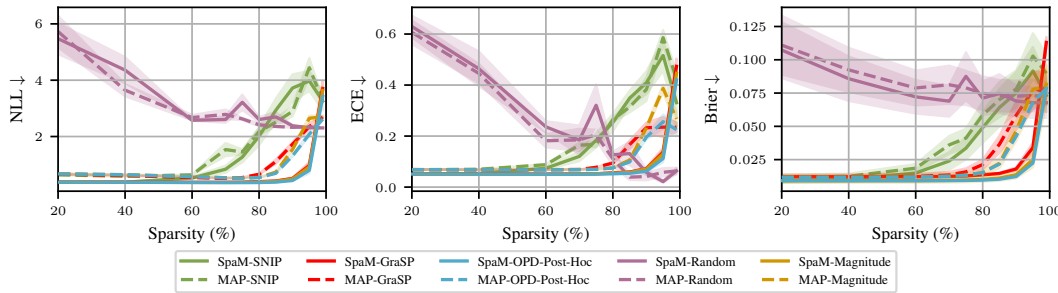

Figure 3: Uncertainty estimation with pruned ResNets on CIFAR-10. We see that SpaM improves uncertainty estimation in terms of NLL, ECE, and Brier score for many pruning criteria and that our OPD criterion outperforms the other criteria, especially at high sparsities.

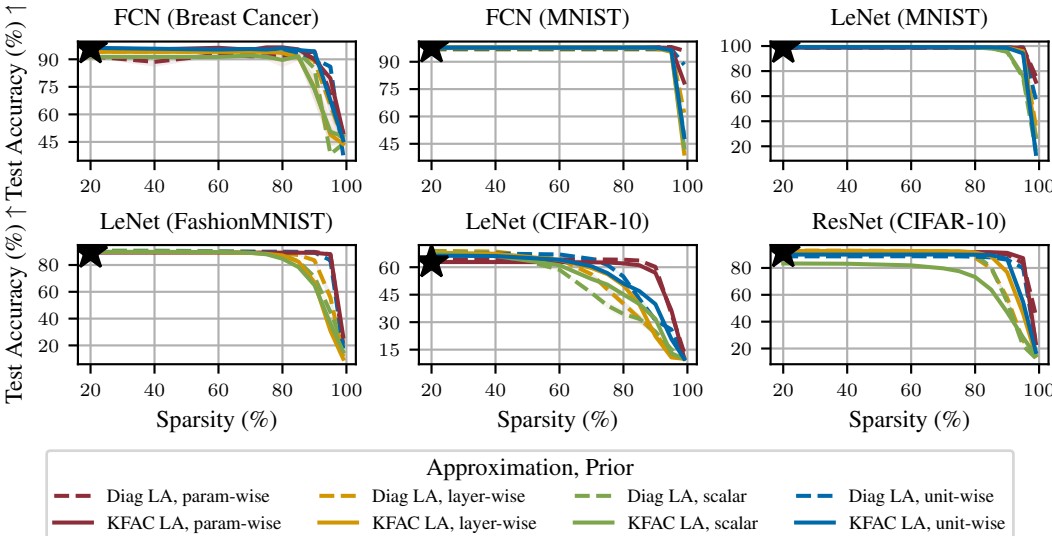

Figure 4: Comparison of different priors and Hessian approximations for SpaM-OPD unstructured pruning. The unit-wise and parameter-wise priors show better performance at high sparsity levels, with the parameter-wise one bridging the gap between Diag and KFAC LA.

**Fine-tuning.** We see in Figure B4 in the appendix that some of this performance difference can be remedied by costly fine-tuning of the networks after pruning, which however still does not allow the other methods to reach the full SpaM-OPD performance. Interestingly, in the case of OPD, this does not further improve its already near-optimal performance.

**Online pruning.** Figure B5 in the appendix shows that our online version of SpaM, which uses the marginal likelihood and OPD during training to iteratively prune the network, reaches comparable performance levels to the post-hoc version, thus offering a computationally even more convenient way to effectively sparsify neural networks.

**Uncertainty estimation.** Given that SpaM is a Bayesian method, it does not only offer high predictive accuracies but also calibrated uncertainty estimates. Indeed, we see in Figure 3 that the trends we have seen for accuracy also apply for negative log-likelihood, expected calibration error, and the Brier score. Again, SpaM improves the uncertainty estimates over MAP training, OPD outperforms most other criteria, and we achieve well-calibrated models up until very high sparsity levels. Note that the random baseline also achieves a low ECE at high sparsity levels because it essentially reverts to random guessing, which is a known weakness of the ECE metric [56].

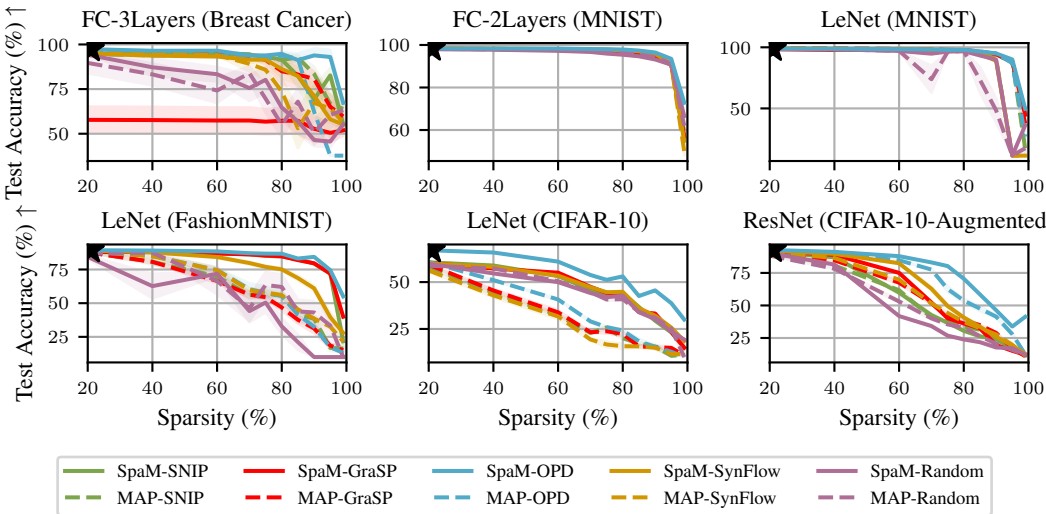

Figure 5: Similarly to unstructured pruning, we see in this experiment on structured pruning that SpaM (using a unit-wise prior) improves performance over MAP and that OPD mostly outperforms other pruning criteria, especially at higher sparsity levels. The black stars reflect the performance of the unpruned models.

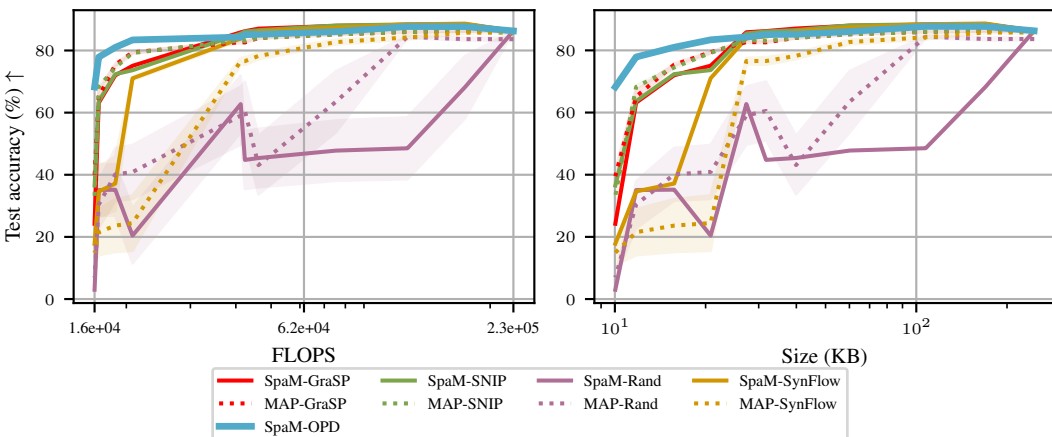

Figure 6: Structured pruning with LeNet on FashionMNIST, using unit-wise priors. We see that our SpaM-OPD dominates the Pareto frontier, in terms of predictive performance as a function of computational time and memory cost, and is particularly competitive at lower costs.

## 5.2 Influence of Priors on Sparsifiability

To understand the influence of the prior and Hessian approximation on performance in our proposed SpaM-OPD approach, we compare diagonal and KFAC approximations with scalar, layer-wise, unit-wise, and parameter-wise priors. Note regarding the latter two, that in this work, we are the first to implement them for the KFAC approximation, thus contributing to the general framework of Laplace-approximated BNNs [14], independent of the pruning use case.

We see in Figure 4 that our newly introduced unit-wise and parameter-wise priors for KFAC indeed outperform the others, especially at high sparsities. When comparing KFAC to the diagonal approximation, we see that KFAC often leads to slightly better performance at lower sparsity levels. However, we also see that the relatively simple choice of parameter-wise prior and diagonal Hessian approximation, as used in our previous experiments above, is a strong baseline across the board and can be recommended as a safe default option for unstructured pruning. Note that the unit-wise priors

can be especially useful for structured pruning, as we will see in the following experiment. More detailed prior comparisons can be found in Appendix B.3.

### 5.3 SpaM Extends to Structured Sparsification

Here, we study the effect of SpaM and OPD in the more challenging setting of eliminating entire network structures, such as convolutional kernels. Studying different network architectures, we aim to generalize our unstructured pruning approach to the setting of structured pruning, where the structures can be freely defined depending on the use case.

Encouragingly, we see in Figure 5 that our findings from the unstructured case transfer qualitatively also to the structured case, with SpaM-OPD outperforming the baselines at high sparsities. Crucially, while the sparsity patterns generated by unstructured pruning are more difficult to translate into computational benefits, structured pruning directly leads to computational savings on standard GPUs (see also Figure B11 in the appendix). We see in Figure 6 that SpaM-OPD dominates the Pareto frontier of the tradeoff between performance and computational cost at high sparsities (i.e., low costs), yielding 10x–20x savings in FLOPS and memory consumption with only minimal deterioration in performance. This positions our proposed framework as a potentially promising approach for the deployment of AI models in resource-constrained environments.

## 6 Conclusion

We have shown that the Bayesian marginal likelihood, with its associated *Occam's razor* effect, can be used *during training* to select neural network models that are *inherently more sparsifiable*. Crucially, we have shown that this sparsifiability *extends across different pruning criteria* and enables *large gains in performance and uncertainty estimation*, especially at *high sparsity levels*. Conveniently, the computations needed for the marginal likelihood estimation using the Laplace approximation can be re-used to define a *novel pruning criterion called OPD*, which outperforms many existing (more expensive) criteria in our experiments. We have also presented *guidelines for choosing priors* within our framework and have shown that even in the challenging setting of *structured pruning*, our proposed SpaM approach can yield up to *20x savings in computational time and memory*, with only small reductions in performance. Our work thus offers a promising path towards pruning large AI models at high sparsity levels for deployment on resource-constrained devices.

**Limitations.** Our approach naturally inherits some limitations of the Laplace approximation, for instance, the fact that it only captures the local geometry of a single posterior mode or potential numerical instabilities in the Hessian computations when used with low-precision weights. Moreover, it accrues an additional computational cost compared to MAP training, which is then, however, amortized by the computational savings during the deployment of the sparsified model.

### Acknowledgments

AI acknowledges funding through a Max Planck ETH Center for Learning Systems (CLS) fellowship. VF was supported by a Branco Weiss Fellowship.

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

# A Proof for Diagonal Prior in a Kronecker-factored Eigenbasis

**Proposition A.1** (Diagonal Prior in KFAC Eigenbasis). *Considering the Frobenius norm, the optimal diagonal perturbation of the KFAC eigenvalues $\mathbf{\Lambda}_A \otimes \mathbf{\Lambda}_B$ to add a diagonal prior precision is given by $\mathbf{\Lambda}_A \otimes \mathbf{\Lambda}_B + \hat{\boldsymbol{\delta}}$ with $\mathrm{mat}(\hat{\boldsymbol{\delta}}) = (\mathbf{Q}_G^{\mathsf{T}})^2 \mathrm{mat}(\boldsymbol{\delta})\mathbf{Q}_A^2$ where the square is element-wise and $\mathrm{mat}(\cdot)$ reshapes the vector to match the parameter shape used in KFAC. Thus, it can be computed efficiently without computing a Kronecker product.*

*Proof.* We prove this result in two steps. First, we show what the optimum looks like in terms of the Frobenius norm. Second, we show how to simplify the results to enable efficient computation without computing Kronecker products. We have a KFAC Hessian approximation $\mathbf{A} \otimes \mathbf{B}$ with $\mathbf{A} \in \mathbb{R}^{D_{\mathrm{in}} \times D_{\mathrm{in}}}$ and $\mathbf{B} \in \mathbb{R}^{D_{\mathrm{out}} \times D_{\mathrm{out}}}$ where the dimensionalities $D.$ depend on the layer type [18]. In the case of a fully-connected layer, these are simply the dimensionality of the in- and output hidden representation. The same layer will have $D_{\mathrm{in}} \times D_{\mathrm{out}}$ parameters and thus the corresponding diagonal prior precision is given by $\boldsymbol{\delta} \in \mathbb{R}^{D_{\mathrm{in}} D_{\mathrm{out}}}$. For the Laplace approximation, the eigendecomposition of individual Kronecker factors is already computed as $\mathbf{A} = \mathbf{Q}_A \mathbf{\Lambda}_A \mathbf{Q}_A^{\mathsf{T}}$ and similarly for $\mathbf{G}$ as shown in Equation (5). Recall also that $\mathrm{diag}(\cdot)$ turns a vector into a diagonal matrix and extracts the diagonal entries of a matrix into a vector. We are interested in the Frobenius-optimal diagonal perturbation of the eigenvalues so as to maintain the efficiency structure of the KFAC, and, thus, the downstream Laplace approximation:

$$\arg\min_{\hat{\boldsymbol{\delta}}} \|(\mathbf{Q}_A \otimes \mathbf{Q}_G)(\mathbf{\Lambda}_A \otimes \mathbf{\Lambda}_G + \mathrm{diag}(\hat{\boldsymbol{\delta}}))(\mathbf{Q}_A^{\mathsf{T}} \otimes \mathbf{Q}_G^{\mathsf{T}})$$

$$- (\mathbf{Q}_A \otimes \mathbf{Q}_G)(\mathbf{\Lambda}_A \otimes \mathbf{\Lambda}_G)(\mathbf{Q}_A^{\mathsf{T}} \otimes \mathbf{Q}_G^{\mathsf{T}}) + \mathrm{diag}(\boldsymbol{\delta})\|_F^2$$

$$= \arg\min_{\hat{\boldsymbol{\delta}}} \|\mathbf{\Lambda}_A \otimes \mathbf{\Lambda}_G + \mathrm{diag}(\hat{\boldsymbol{\delta}}) - \mathbf{\Lambda}_A \otimes \mathbf{\Lambda}_G + (\mathbf{Q}_A^{\mathsf{T}} \otimes \mathbf{Q}_G^{\mathsf{T}})\mathrm{diag}(\boldsymbol{\delta})(\mathbf{Q}_A \otimes \mathbf{Q}_G)\|_F^2$$

$$= \mathrm{diag}((\mathbf{Q}_A^{\mathsf{T}} \otimes \mathbf{Q}_G^{\mathsf{T}})\mathrm{diag}(\boldsymbol{\delta})(\mathbf{Q}_A \otimes \mathbf{Q}_G)),$$

where we first multiplied the orthogonal bases from left and right and then realized that the values of $\hat{\boldsymbol{\delta}}$ need to be set to the entries of the prior $\boldsymbol{\delta}$ projected into the basis.

Naïvely, computing the optimum of $\hat{\boldsymbol{\delta}}$ would require expanding the Kronecker product above and lead to a potentially intractable complexity of $\mathcal{O}(D_{\mathrm{in}}^2 D_{\mathrm{out}}^2)$. However, it is possible to simplify it further to maintain efficient computation: For simplicity, consider the case without Kronecker factorization. We have

$$\mathrm{diag}(\mathbf{Q}^{\mathsf{T}}\mathrm{diag}(\mathbf{d})\mathbf{Q}) = (\mathbf{Q}^{\mathsf{T}} \circ \mathbf{Q}^{\mathsf{T}})\mathbf{d},$$

where $\circ$ is the element-wise Hadamard product. So we can express the diagonal of the matrix-matrix product as a matrix-vector product with the diagonal $\mathbf{d}$ as the vector. In the Kronecker-factored case, we need just one more simplification:

$$\mathrm{diag}((\mathbf{Q}_A^{\mathsf{T}} \otimes \mathbf{Q}_G^{\mathsf{T}})\mathrm{diag}(\boldsymbol{\delta})(\mathbf{Q}_A \otimes \mathbf{Q}_G)) = ((\mathbf{Q}_A^{\mathsf{T}} \otimes \mathbf{Q}_G^{\mathsf{T}}) \circ (\mathbf{Q}_A^{\mathsf{T}} \otimes \mathbf{Q}_G^{\mathsf{T}}))\boldsymbol{\delta}$$

$$= ((\mathbf{Q}_A^{\mathsf{T}} \circ \mathbf{Q}_A^{\mathsf{T}}) \otimes (\mathbf{Q}_G^{\mathsf{T}} \circ \mathbf{Q}_G^{\mathsf{T}}))\boldsymbol{\delta}$$

$$= \mathrm{vec}((\mathbf{Q}_G^{\mathsf{T}} \circ \mathbf{Q}_G^{\mathsf{T}})\mathrm{mat}(\boldsymbol{\delta})(\mathbf{Q}_A \circ \mathbf{Q}_A))$$

$$= \mathrm{vec}(\mathbf{Q}_G^{\mathsf{T}})^2 \mathrm{mat}(\boldsymbol{\delta})\mathbf{Q}_A^2,$$

where we have used the mixed-product property of the Kronecker product and the properties for multiplying a Kronecker-product with a vector. The $\mathrm{vec}$ operator "flattens" a matrix, that is, turns a $D_{\mathrm{out}} \times D_{\mathrm{in}}$ matrix into a $D_{\mathrm{out}} D_{\mathrm{in}}$ vector, and $\mathrm{mat}$ does the opposite. The final approximation $\hat{\boldsymbol{\delta}}$ can be computed efficiently in $\mathcal{O}(D_{\mathrm{in}}^2 + D_{\mathrm{out}}^2)$. $\qquad\square$

# B Additional Results

## B.1 Baseline training

Figure B1 illustrates that both MAP and SPAM achieve similar levels of performance throughout the training process. This observation underscores that SPAM's enhanced sparsifiability is achieved without compromising the unpruned performance. Furthermore, the comparable unpruned accuracies

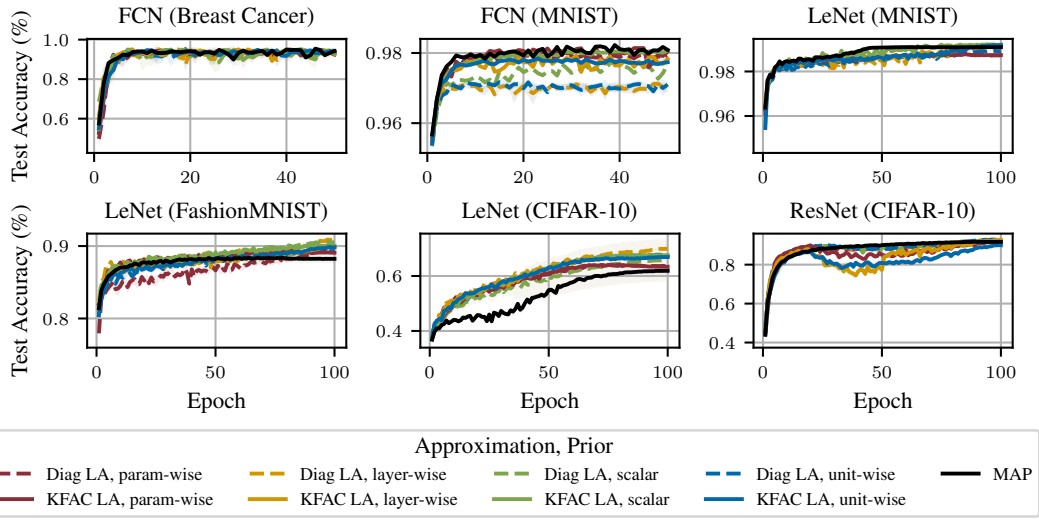

Figure B1: Training curves for MAP and SpaM training with different priors and Hessian approximations. We see that all methods achieve a similar performance by the end of training.

of SPAM and MAP models indicate that SPAM's sparsifiability benefits are not merely a result of higher baseline accuracies, but rather a distinct advantage offered by the SPAM methodology. The sparsification methods are performed on these models in a way that once the model is trained for a specific seed, we copy it and use it to perform the different sparsification methods; we repeat the steps for a minimum of 4 different seeds, ensuring the robustness of our findings. In addition, the model trained with SpaM only uses a single forward pass over the pruned architecture during inference, thus guaranteeing a fair comparison with the baselines. The posterior is solely used to estimate the marginal likelihood during SpaM training.

## B.2 Tables

In tables B1 and B2, we present our results comparing different methods using MAP and SpaM with various priors. Notably, SpaM with Diag LA and parameter-wise priors significantly outperforms MAP and other SpaM variants at high sparsity levels.

## B.3 Prior effects

Figure B2 and Figure B3 illustrate our findings when applying SpaM with various priors for both OPD and GraSP. Notably, Diag LA, using parameter-wise priors, excels in high-sparsity scenarios, even with complex models and datasets like ResNets. Furthermore, for MLPmixer, we observe that SpaM variants, employing parameter-wise priors and layerwise approaches, preserve baseline accuracy even at extreme sparsities of 99%.

## B.4 Unit-wise and Parameter-wise KFAC for GraSP

As shown in Section 5, networks trained using SpaM and parameter-wise priors were able to maintain a high accuracy at challenging sparsity levels up to 99%. Moreover, parameter-wise KFAC and unit-wise priors showed high performance for the OPD pruning approach. We show in Figure B3 that the combination of SpaM and these priors with the GraSP criterion yield qualitatively similar performance rankings as with our OPD criterion.

## B.5 One Shot Efficiency

As seen in Figure B4, our proposed post-hoc pruning criterion, OPD, consistently demonstrates stable performance across diverse model architectures and datasets, achieving significant sparsity levels without the need for fine-tuning. It seamlessly operates either post-training or with pre-trained models, providing a highly flexible and versatile solution.

Table B1: Comparison of pruning accuracies of SpaM training with different pruning criteria, Hessian approximations, and priors for post-hoc pruning LeNet on MNIST.

| Criterion | Approximation | Sparsity (%) Prior | 20 | 40 | 60 | 70 | 75 | 80 | 85 | 90 | 95 | 99 |
|---|---|---|---|---|---|---|---|---|---|---|---|---|
| GraSP | Diag | parameter-wise | 98.75 | 98.75 | 98.74 | 98.75 | 98.75 | 98.75 | 98.75 | **98.74** | **98.72** | 52.19 |
| | | layerwise | 99.08 | 99.09 | 99.10 | 98.96 | 98.50 | 97.79 | 93.50 | 78.93 | 63.71 | 16.67 |
| | | scalar | 99.11 | 99.11 | 99.12 | 98.84 | 98.41 | 97.74 | 87.67 | 57.99 | 44.60 | 15.73 |
| | KFAC | layerwise | **99.25** | **99.25** | **99.26** | 99.01 | 98.74 | 97.86 | 91.48 | 82.78 | 50.75 | 13.48 |
| | | scalar | 99.24 | 99.24 | 99.24 | 99.02 | 98.58 | 98.02 | 92.03 | 73.64 | 42.18 | 12.23 |
| | MAP | MAP | 99.01 | 99.01 | 99.01 | 98.99 | 98.91 | 98.75 | 98.28 | 96.10 | 77.30 | 20.43 |
| SNIP | Diag | parameter-wise | 98.73 | 98.73 | 98.68 | 97.95 | 95.70 | 83.90 | 57.55 | 20.47 | 13.00 | 9.10 |
| | | layerwise | 99.08 | 98.34 | 46.42 | 17.69 | 21.89 | 14.61 | 14.47 | 13.50 | 11.94 | 9.83 |
| | | scalar | 99.11 | 99.11 | 98.56 | 91.82 | 84.70 | 62.25 | 34.56 | 10.75 | 16.28 | 16.70 |
| | KFAC | layerwise | **99.25** | 99.20 | 79.59 | 27.70 | 43.76 | 23.85 | 10.28 | 16.19 | 19.90 | 10.27 |
| | | scalar | 99.24 | 99.24 | 98.56 | 94.82 | 70.25 | 48.28 | 27.02 | 26.88 | 25.95 | 9.86 |
| | MAP | MAP | 99.01 | 99.01 | 98.95 | 98.31 | 97.21 | 94.26 | 87.19 | 65.42 | 25.15 | 12.56 |
| OPD | Diag | parameter-wise | 98.72 | 98.72 | 98.72 | 98.72 | 98.72 | 98.72 | 98.72 | 98.72 | **98.72** | **75.92** |
| | | layerwise | 99.08 | 99.09 | 99.08 | 99.07 | 99.04 | 98.98 | 98.84 | 98.40 | 94.71 | 36.25 |
| | | scalar | 99.11 | 99.11 | 99.10 | 99.11 | 99.06 | 98.89 | 98.28 | 95.32 | 74.78 | 16.12 |
| | KFAC | layerwise | **99.25** | **99.25** | 99.24 | **99.23** | **99.18** | **99.11** | **98.95** | 98.60 | 95.90 | 28.16 |
| | | scalar | 99.24 | 99.24 | 99.24 | 99.16 | 99.10 | 98.90 | 97.97 | 95.88 | 76.11 | 27.38 |
| | MAP | MAP | 99.01 | 99.01 | 99.03 | 98.99 | 98.95 | 98.90 | 98.71 | 98.17 | 92.82 | 27.19 |
| Magnitude | Diag | parameter-wise | 98.72 | 98.72 | 98.72 | 98.70 | 98.69 | 98.67 | 98.65 | 98.59 | 98.03 | 38.61 |
| | | layerwise | 99.08 | 99.09 | 99.08 | 99.06 | 99.01 | 98.92 | 98.46 | 94.20 | 39.86 | 10.19 |
| | | scalar | 99.11 | 99.11 | 99.09 | 99.12 | 99.07 | 98.98 | 98.48 | 95.69 | 74.30 | 13.53 |
| | KFAC | layerwise | **99.25** | **99.25** | 99.22 | 99.18 | 99.08 | 98.95 | 98.38 | 91.93 | 28.52 | 9.80 |
| | | scalar | 99.24 | 99.24 | 99.20 | 99.14 | 99.04 | 98.92 | 98.62 | 97.08 | 84.66 | 22.21 |
| | MAP | MAP | 99.01 | 99.01 | 98.99 | 98.96 | 98.93 | 98.85 | 98.57 | 97.69 | 88.82 | 15.42 |
| Random | Diag | parameter-wise | 55.88 | 25.95 | 11.15 | 10.88 | 11.13 | 10.56 | 11.88 | 11.35 | 9.95 | 9.81 |
| | | layerwise | 78.86 | 17.47 | 22.35 | 14.23 | 12.06 | 11.80 | 10.18 | 12.82 | 9.35 | 9.80 |
| | | scalar | 88.75 | 60.17 | 25.35 | 15.98 | 14.26 | 11.91 | 8.63 | 9.74 | 9.05 | 9.80 |
| | KFAC | layerwise | 89.90 | 18.70 | 20.36 | 14.13 | 14.56 | 12.61 | 9.72 | 12.00 | 8.93 | 9.80 |
| | | scalar | 90.46 | 34.14 | 19.98 | 12.72 | 8.49 | 11.40 | 10.08 | 10.54 | 9.74 | 9.80 |
| | MAP | MAP | 79.03 | 43.25 | 22.86 | 9.68 | 10.34 | 9.96 | 11.50 | 9.50 | 10.85 | 9.80 |

## B.6 Online pruning.

Figure B5 shows that our online version of SpaM, which uses the marginal likelihood and OPD during training to iteratively prune the network, reaches comparable performance levels to the post-hoc version, thus offering a computationally even more convenient way to effectively sparsify neural networks. In the online approach, we prune jointly during SpaM training to reach a target sparsity; we perform the sparsity updates (dynamic masking) based on the marginal likelihood optimization parameters we refer to as *n_epochs_burnin* and *marglik_frequency*. Here, *n_epochs_burnin* specifies when we start the marginal likelihood optimization and *marglik_frequency* specifies after how many epochs we update the estimate. Pruning occurs only after a new marginal likelihood calculation, and the sparsity percentage is adjusted incrementally to reach the target by the training's end. The curve of OPD-Online reflects training progress (x-axis explicitly epochs / reached sparsity), which explains the curve of LeNet on CIFAR-10 that is still converging while pruning. At the same time, the one-shot post-training approach reflects converged models copied and pruned at different sparsity levels.

## B.7 Comparing SpaM to L1 regularization

We conducted experiments using L1 regularization, which is well-known for inducing sparsity in neural networks. To optimize performance, we performed an extensive search for the appropriate L1 regularization coefficient and applied various strengths during training.

We found that SpaM consistently outperforms L1 regularization, achieving much higher levels of sparsity while maintaining the network's predictive performance as shown in Figure B6. L1 regularization, while effective at inducing sparsity, often proved too aggressive, leading to networks that were excessively pruned, negatively affecting performance. In contrast, our method offers a key

Table B2: Comparison of pruning accuracies of SpaM training with different pruning criteria, Hessian approximations, and priors for post-hoc pruning MLP-Mixer (2 blocks) on MNIST.

| Method | Approximation | Sparsity (%) Prior | 20 | 40 | 60 | 70 | 75 | 80 | 85 | 90 | 95 | 99 |
|---|---|---|---|---|---|---|---|---|---|---|---|---|
| GraSP | Diag | parameter-wise | **98.51** | **98.51** | **98.51** | **98.51** | **98.51** | 98.51 | **98.52** | 98.51 | **98.52** | **98.50** |
| | | layerwise | 97.71 | 97.71 | 97.71 | 97.71 | 97.71 | 97.71 | 97.71 | 97.71 | 97.71 | 90.35 |
| | | scalar | 97.91 | 97.91 | 97.91 | 97.91 | 97.91 | 97.91 | 97.91 | 97.91 | 97.91 | 92.54 |
| | KFAC | parameter-wise | 98.10 | 98.10 | 98.10 | 98.10 | 98.10 | 98.10 | 98.10 | 98.10 | 98.11 | 95.45 |
| | | layerwise | 98.16 | 98.16 | 98.16 | 98.16 | 98.16 | 98.16 | 98.16 | 98.15 | 97.92 | 57.14 |
| | | scalar | 98.29 | 98.29 | 98.29 | 98.29 | 98.29 | 98.29 | 98.29 | 98.29 | 98.21 | 64.88 |
| | MAP | MAP | 98.41 | 98.41 | 98.38 | 98.38 | 98.33 | 98.35 | 98.17 | 97.38 | 89.43 | 38.89 |
| SNIP | Diag | parameter-wise | **98.51** | **98.51** | **98.51** | **98.51** | **98.51** | **98.52** | 98.49 | 97.70 | 83.76 | 20.00 |
| | | layerwise | 97.71 | 97.71 | 97.71 | 97.71 | 97.71 | 97.71 | 97.71 | 97.71 | 97.71 | 12.41 |
| | | scalar | 97.91 | 97.91 | 97.91 | 97.91 | 97.91 | 97.91 | 97.91 | 97.91 | 97.91 | 75.84 |
| | KFAC | parameter-wise | 98.10 | 98.10 | 98.10 | 98.10 | 98.10 | 98.10 | 98.10 | 98.10 | 97.84 | 32.84 |
| | | layerwise | 98.16 | 98.16 | 98.16 | 98.16 | 98.15 | 98.15 | 98.14 | 98.10 | 92.04 | 28.16 |
| | | scalar | 98.29 | 98.29 | 98.29 | 98.29 | 98.29 | 98.29 | 98.29 | 98.29 | 97.89 | 54.69 |
| | MAP | MAP | 98.38 | 98.36 | 98.34 | 98.22 | 98.02 | 97.44 | 96.04 | 92.22 | 81.42 | 35.36 |
| OPD | Diag | parameter-wise | **98.51** | **98.51** | **98.51** | **98.51** | **98.51** | 98.51 | **98.52** | **98.52** | 98.51 | **98.50** |
| | | layerwise | 97.71 | 97.71 | 97.71 | 97.71 | 97.71 | 97.71 | 97.71 | 97.71 | 97.71 | 96.06 |
| | | scalar | 97.91 | 97.91 | 97.91 | 97.91 | 97.91 | 97.91 | 97.91 | 97.91 | 97.91 | 96.47 |
| | KFAC | parameter-wise | 98.10 | 98.10 | 98.10 | 98.10 | 98.10 | 98.10 | 98.10 | 98.10 | 98.10 | 96.97 |
| | | layerwise | 98.16 | 98.16 | 98.16 | 98.16 | 98.16 | 98.16 | 98.16 | 98.15 | 97.84 | 86.81 |
| | | scalar | 98.29 | 98.29 | 98.29 | 98.29 | 98.29 | 98.29 | 98.29 | 98.29 | 98.23 | 84.91 |
| | MAP | MAP | 98.38 | 98.39 | 98.38 | 98.35 | 98.34 | 98.32 | 98.20 | 97.72 | 94.26 | 57.66 |
| Magnitude | Diag | parameter-wise | **98.51** | **98.51** | **98.51** | **98.51** | **98.51** | **98.52** | **98.52** | 98.51 | **98.52** | 98.48 |
| | | layerwise | 97.71 | 97.71 | 97.71 | 97.71 | 97.71 | 97.71 | 97.71 | 97.71 | 97.70 | 76.89 |
| | | scalar | 97.91 | 97.91 | 97.91 | 97.91 | 97.91 | 97.91 | 97.91 | 97.91 | 97.91 | 96.24 |
| | KFAC | parameter-wise | 98.10 | 98.10 | 98.10 | 98.10 | 98.10 | 98.10 | 98.10 | 98.10 | 98.10 | 94.22 |
| | | layerwise | 98.16 | 98.16 | 98.16 | 98.16 | 98.16 | 98.16 | 98.16 | 98.14 | 97.89 | 36.31 |
| | | scalar | 98.29 | 98.29 | 98.29 | 98.29 | 98.29 | 98.29 | 98.29 | 98.29 | 98.24 | 82.57 |
| | MAP | MAP | 98.38 | 98.39 | 98.38 | 98.36 | 98.34 | 98.30 | 98.16 | 97.77 | 93.56 | 53.16 |
| Random | Diag | parameter-wise | 90.09 | 62.13 | 39.76 | 31.51 | 19.52 | 22.50 | 18.93 | 14.84 | 15.57 | 11.81 |
| | | layerwise | 83.18 | 53.42 | 32.12 | 28.44 | 22.51 | 22.82 | 18.88 | 17.00 | 12.95 | 11.02 |
| | | scalar | 85.66 | 57.33 | 37.74 | 25.43 | 23.45 | 20.28 | 22.31 | 16.64 | 14.89 | 9.85 |
| | KFAC | parameter-wise | 85.95 | 58.41 | 39.27 | 27.96 | 24.64 | 21.94 | 18.72 | 17.04 | 13.66 | 9.37 |
| | | layerwise | 87.00 | 58.40 | 41.93 | 33.39 | 30.62 | 27.91 | 20.77 | 21.02 | 14.07 | 10.49 |
| | | scalar | 90.16 | 64.54 | 40.36 | 34.11 | 30.63 | 28.92 | 17.88 | 18.35 | 13.82 | 11.47 |
| | MAP | MAP | 97.26 | 87.54 | 65.94 | 52.32 | 47.22 | 44.12 | 40.72 | 22.46 | 16.59 | 8.87 |

Table B3: NLL of unstructured pruned ResNets on CIFAR-10. The best training method for each pruning criterion is highlighted in green, showing that SpaM improves performance over MAP for most criteria. The best performances (lowest NLL) overall at each sparsity level are shown in **bold**, showing that our OPD criterion outperforms the others at most sparsity levels.

| Criterion | Training | Sparsity (%) | | | | | | |
|---|---|---|---|---|---|---|---|---|
| | | 70 | 75 | 80 | 85 | 90 | 95 | 99 |
| OPD | MAP | 0.53 ± 0.0013 | 0.52 ± 0.0011 | 0.54 ± 0.0011 | 0.69 ± 0.0036 | 1.31 ± 0.0086 | 2.08 ± 0.0190 | 2.62 ± 0.0106 |
| | SpaM | **0.36 ± 0.0016** | **0.36 ± 0.0016** | **0.37 ± 0.0014** | **0.38 ± 0.0022** | **0.44 ± 0.0056** | **0.80 ± 0.0270** | 3.43 ± 0.0179 |
| GraSP | MAP | 0.51 ± 0.0008 | 0.54 ± 0.0032 | 0.66 ± 0.0046 | 1.11 ± 0.0195 | 1.73 ± 0.0193 | 2.35 ± 0.0276 | 2.69 ± 0.0088 |
| | SpaM | 0.37 ± 0.0007 | 0.38 ± 0.0006 | 0.40 ± 0.0015 | 0.42 ± 0.0032 | 0.51 ± 0.0093 | 0.97 ± 0.0317 | 3.71 ± 0.0709 |
| Magnitude | MAP | 0.54 ± 0.0014 | 0.53 ± 0.0011 | 0.55 ± 0.0015 | 0.73 ± 0.0034 | 1.54 ± 0.0098 | 2.65 ± 0.0239 | 2.70 ± 0.0113 |
| | SpaM | 0.37 ± 0.0011 | 0.37 ± 0.0012 | 0.38 ± 0.0016 | 0.41 ± 0.0028 | 0.49 ± 0.0072 | 0.92 ± 0.0320 | 3.63 ± 0.0418 |
| Random | MAP | 2.79 ± 0.0438 | 2.63 ± 0.0089 | 2.42 ± 0.0146 | 2.36 ± 0.0042 | 2.33 ± 0.0037 | 2.34 ± 0.0043 | **2.30 ± 0.0000** |
| | SpaM | 2.60 ± 0.0292 | 3.22 ± 0.0701 | 2.60 ± 0.0230 | 2.70 ± 0.0447 | 2.38 ± 0.0044 | 2.31 ± 0.0009 | 2.31 ± 0.0003 |
| SNIP | MAP | 1.54 ± 0.0595 | 1.45 ± 0.0235 | 2.18 ± 0.0331 | 2.51 ± 0.0350 | 2.90 ± 0.0450 | 4.44 ± 0.0864 | 3.28 ± 0.0203 |
| | SpaM | 0.84 ± 0.0320 | 1.27 ± 0.0474 | 2.01 ± 0.0814 | 2.93 ± 0.1060 | 3.72 ± 0.1244 | 3.97 ± 0.0907 | 3.26 ± 0.0841 |

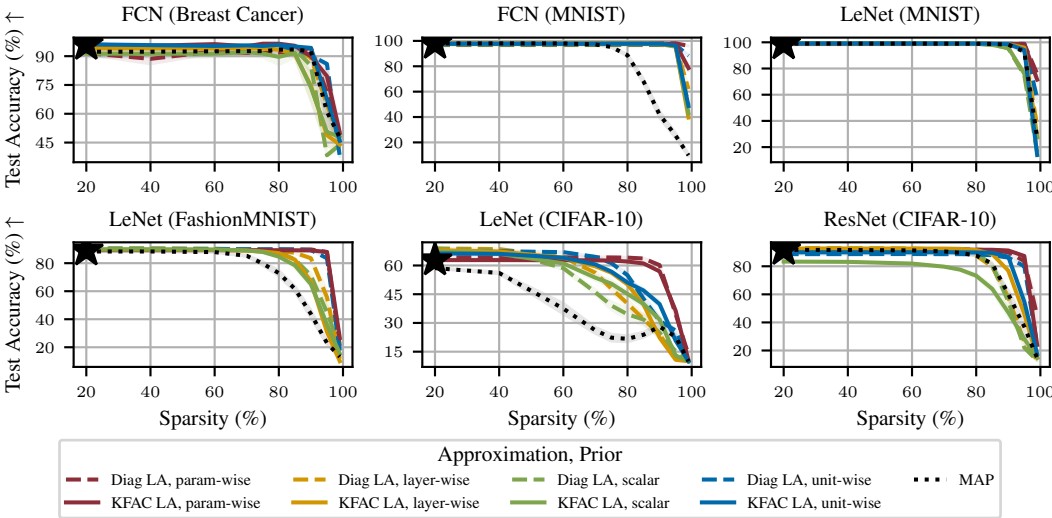

Figure B2: Effect of different priors and Hessian approximations on the sparsification performance with SpaM-OPD. The diagonal approximation with parameter-wise priors is a strong choice, especially at higher sparsities, while the KFAC approximation with layerwise prior yields slightly better performances at lower sparsities.

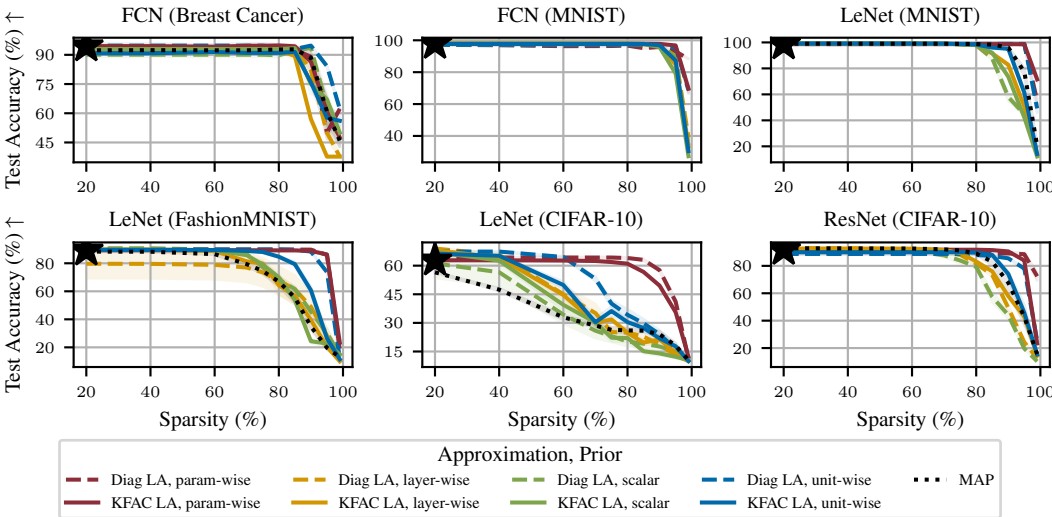

Figure B3: Priors and Hessian approximations for GraSP pruning with SpaM and MAP, including our newer priors (parameter-wise and unit-wise). We see that the effects are qualitatively similar to pruning with OPD.

advantage: it adapts the regularization to each individual weight based on the data, rather than relying on a single global parameter, allowing for more nuanced control over sparsity.

## B.8 Additional Pruning Results

### B.8.1 Wide ResNet

In Figure B7, we demonstrate how SpaM enhances the sparsity performance of Wide ResNet models. This is specifically illustrated in the case of OPD, GraSP, and Magnitude, all while maintaining a low Brier score, ECE, and NLL up to 95% sparsity.

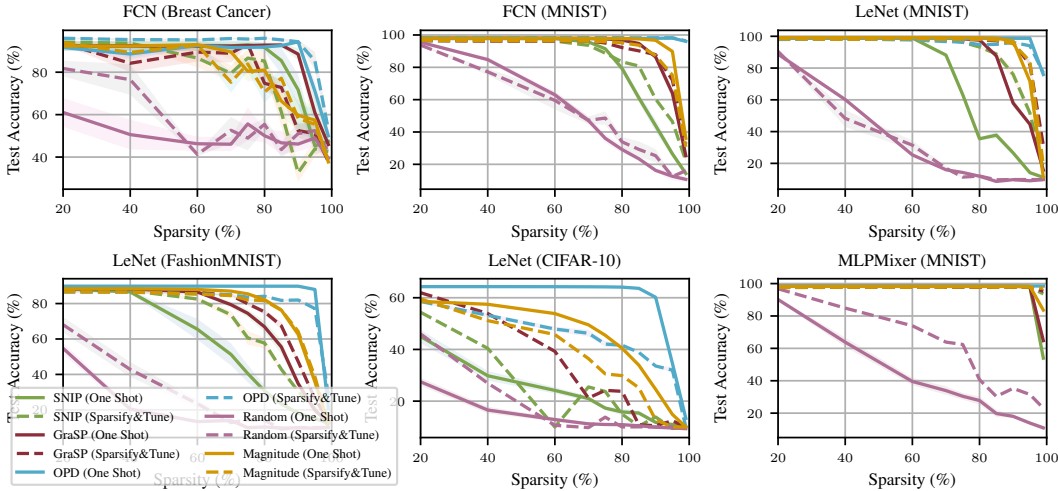

Figure B4: SpaM post-hoc pruning efficiency with optional fine-tuning after the pruning. Unlike other pruning criteria, OPD does not require additional tuning to achieve optimal performance across different architectures and often still outperforms the other fine-tuned methods.

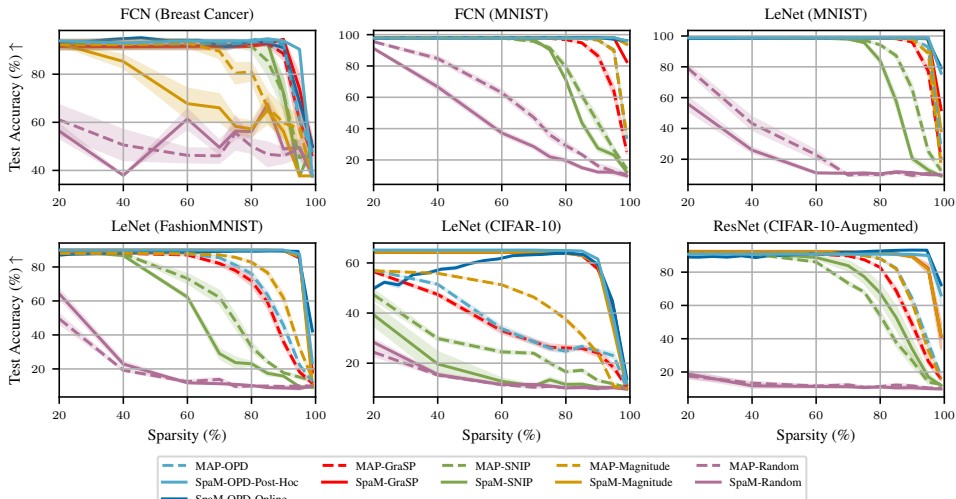

Figure B5: Predictive performance as a function of sparsity level in unstructured pruning. We include our online pruning approach that progressively prunes **a** model during the training compared to the other curves demonstrating the performance of 10 pruned models based on **a converged** baseline. our online pruning approach is often competitive with post-hoc pruning.

### B.8.2 Vision Transformer

Figure B8 demonstrates the impact of SpaM on Unstructured Pruning for a Vision Transformer (ViT) trained on MNIST. These results align with the findings presented in Section 5 with SpaM diagonal LA and parameter-wise priors leveraging the sparsifiability of models using OPD, Magnitude and GraSP, maintaining a test accuracy of 97% for OPD and Magnitude at 95% sparsity compared to an accuracy of lower than 20% under MAP for the same methods. This serves as a proof-of-concept for vision transformers but efficacy has to be verified at a larger scale where such models perform best.

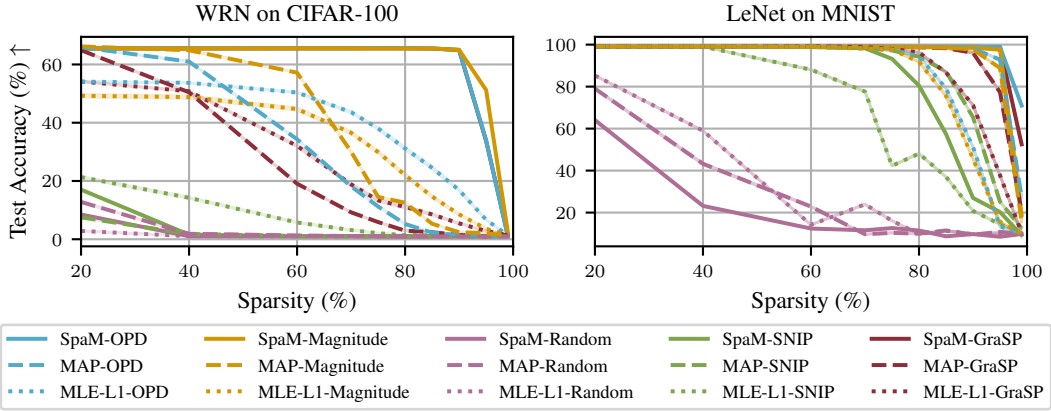

Figure B6: Comparing SpaM and MAP to L1 Regularization. We see that SpaM consistently outperforms L1 regularization.

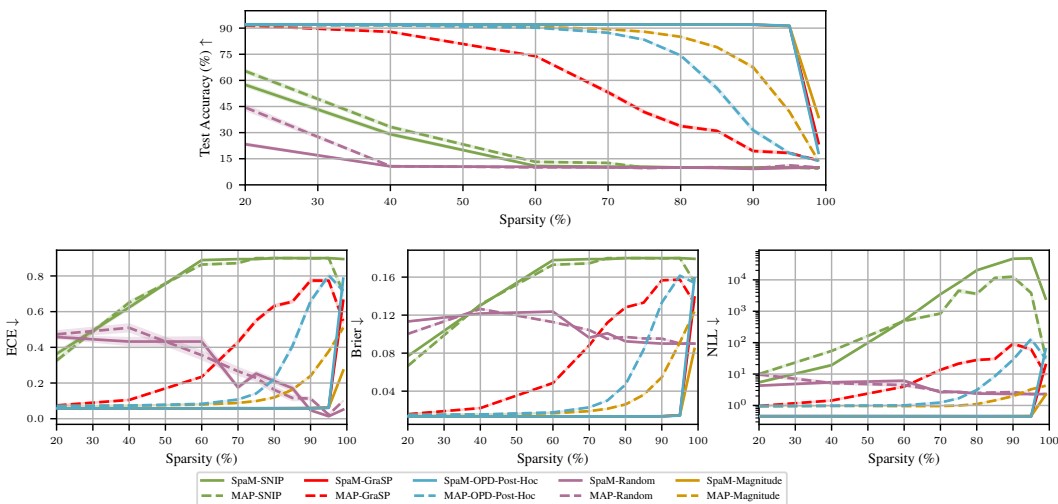

Figure B7: SpaM post-hoc efficiency for Wide ResNet on CIFAR10. Leveraging OPD, GraSP, and Magnitude performance under SpaM in comparison to MAP show superior test accuracy at increased sparsity levels coupled with a low ECE, Brier Score, and NLL.

### B.8.3 GPT-2

We demonstrate the efficacy of OPD on a pre-trained GPT-2 model (124M parameters), fine-tuned for sentiment analysis on the IMDB dataset. To manage computational resources, we limit both the Laplace approximation and SpaM to two steps. Despite this constraint, OPD maintains high predictive performance even at 60% sparsity, as shown in Figure B9. This suggests that extending SpaM optimization with more epochs and a more refined posterior could further enhance performance.

### B.9 Visualization of the Pruning Process

In order to illustrate the structural evolution of the model throughout the different sparsification approaches (unstructured and structured), we provide a sequence of filter bank visualizations that delineate the principal stages of pruning, progressing from the initial dense architecture to the ultimate compact configuration. These visualizations also reveal the influence of parameter-wise and unit-wise priors on the weights (Figure B10).

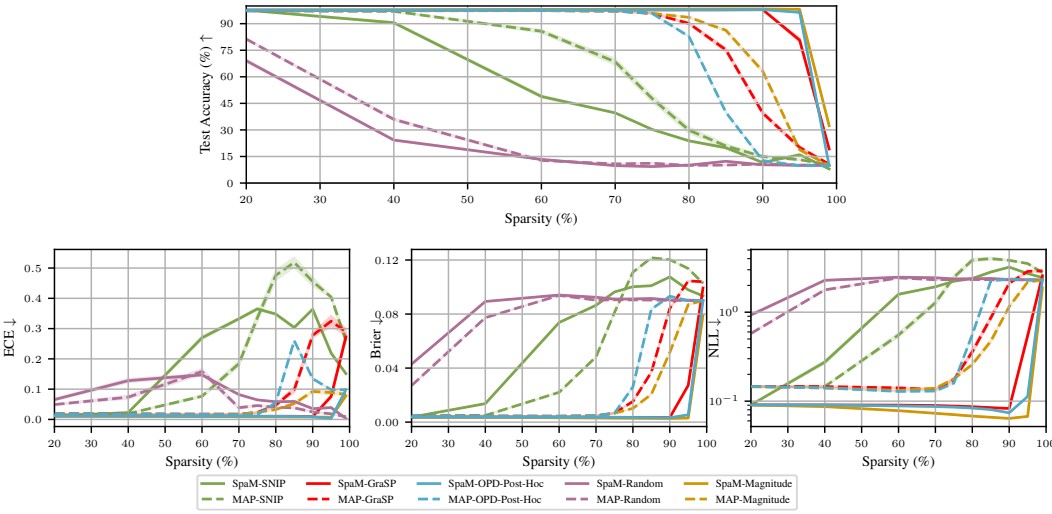

Figure B8: SpaM post-hoc efficiency for ViT on MNIST. Leveraging OPD, GraSP, and Magnitude performance under SpaM in comparison to MAP show superior test accuracy at increased sparsity levels coupled with a low ECE, Brier Score, and NLL.

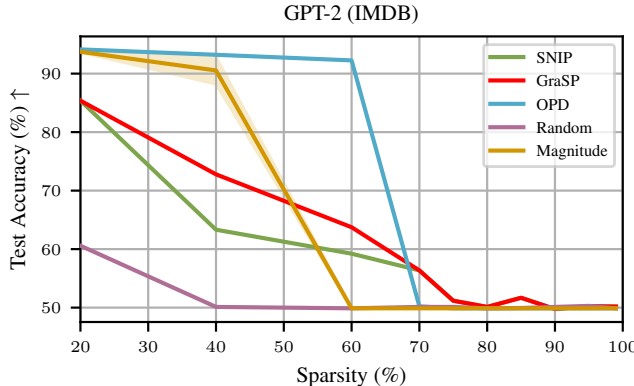

Figure B9: GPT-2 (124M) on IMDB. We tune GPT-2 for sentiment analysis on IMDB datasets. Our results show that OPD maintains significantly higher accuracy than other methods, which degrade towards random classifier performance (50%) at 60% sparsities.

## B.10    Network Compression

In Figures B11 and B12, we demonstrate the efficiency gains achieved by our SpaM-OPD approach. For the fully connected network on the Cancer dataset, it achieves a remarkable reduction of over 20 times in disk size and 24 times in FLOPs while simultaneously maintaining baseline test accuracy. Additionally, it boasts a Brier score of 0.15 and a negative log marginal likelihood (Neg Log MargLik) lower than the original model. These results highlight the effectiveness of SpaM-OPD in achieving significant model compression without compromising performance on key metrics.

## C    Technical Details

### C.1    Resizing and Compression

Post structured pruning, the model may undergo fine-tuning to regain performance. In this process, pruned structures are completely removed from the architecture rather than merely being zeroed out.

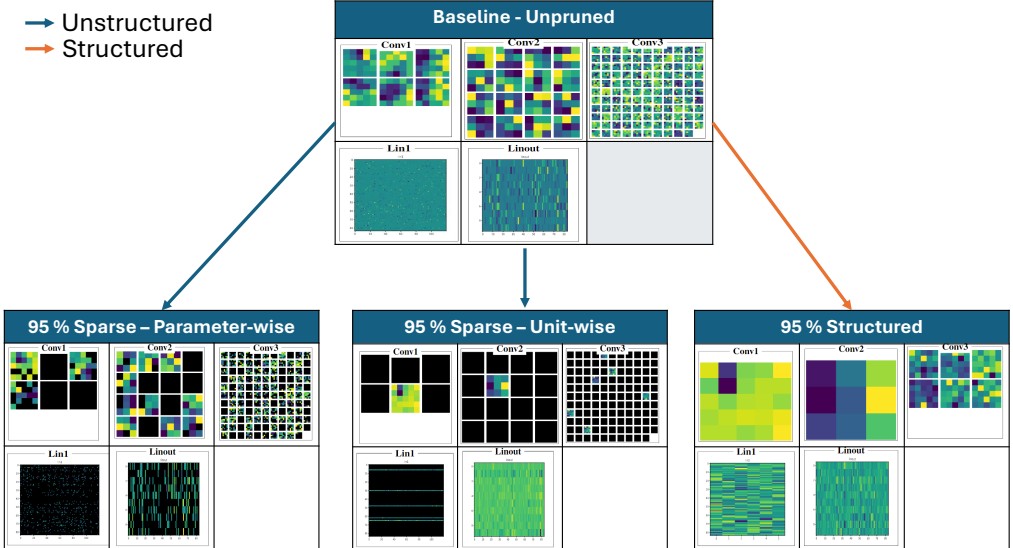

Figure B10: Visualization of the sparsification effect on the model's layers for LeNet on FashionM-NIST sparsified at 95% using both the structured and unstructured approach. Blocks in black refer to masked filters, and columns refer to neurons pruned. We see that for unit-wise priors, a 95% sparsity yields more entire kernels and neurons being masked compared to parameter-wise priors.

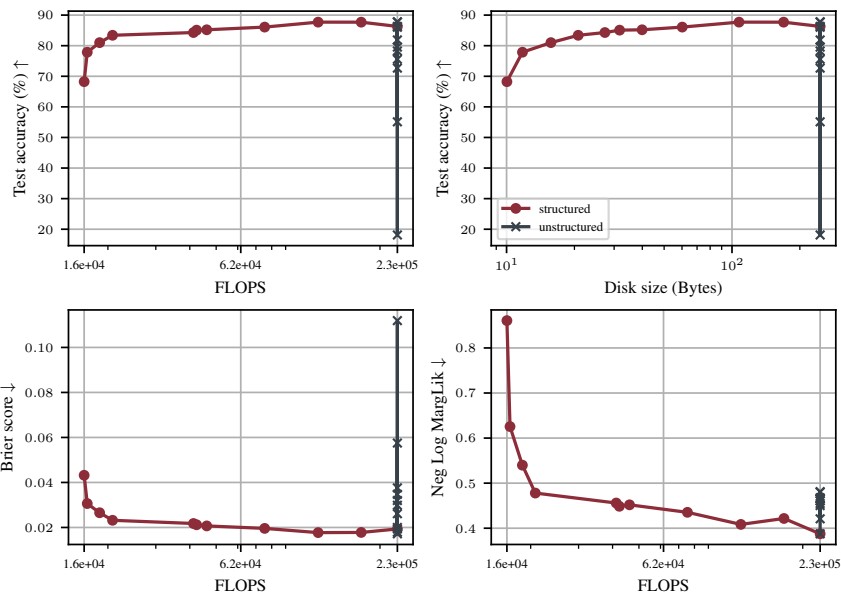

Figure B11: Structured and unstructured pruning of LeNet on FashionMNIST with SpaM-OPD. We see that through structured sparsification, we are able to obtain models that are still performant at a significantly reduced computational and memory cost, while unstructured pruning does not directly translate into computational benefits.

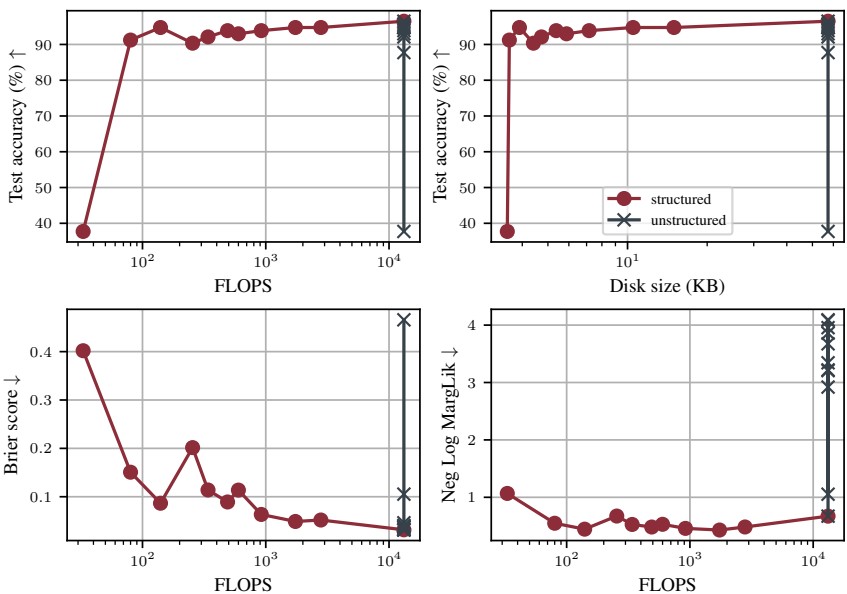

Figure B12: Structured and unstructured pruning of FC on Cancer with SpaM-OPD. We see that through structured sparsification, we are able to obtain models that are still performant at a significantly reduced computational and memory cost. At the same time, unstructured pruning does not directly translate into computational benefits.

This leads to a network with fewer filters in convolutional layers and a reduced number of neurons in fully connected layers, resulting in a leaner and more efficient model.

The process of compaction involves transferring the weights from the pruned model to a newly created, smaller architecture that is aligned with the dimensions of the retained active structures. This results in a denser, storage- and computation-optimized model.

Algorithm 1 summarizes this entire process of structured pruning and model compaction.

This approach transitions the model from a pruned state to a compact and optimized architecture. The final compressed model $M_{\text{compact}}$ not only retains essential predictive capabilities but is also further tuned for performance. The newly configured $M_{\text{compact}}$ is saved with updated parameters, ensuring efficient inference and ease of deployment, especially on resource-constrained edge devices.

The reduced memory footprint and FLOPS of $M_{\text{compact}}$ are particularly beneficial for deployment on edge devices with limited computational resources. When models exceed the hardware limits, aggressive compression techniques like quantization may be required, which can compromise performance. Our method aims to significantly reduce the memory size of the model while minimizing performance trade-offs. The effectiveness of our approach in achieving this balance is explored in Section 5.

## C.2 Pseudocodes

Algorithm 1 outlines our structured pruning procedure, highlighting how we efficiently achieve a simpler model by transferring weights to a smaller one.

# D Experimental Setup

## D.1 Datasets

- *Breast Cancer Wisconsin (Diagnostic) (UCI)*: This dataset, derived from digitized images of fine needle aspirates of breast masses, includes features describing characteristics of cell

---

**Algorithm 1** Structured OPD pruning

---

**Require:** Trained Model $M$, Target Sparsity Threshold $T$
**Ensure:** Compacted Model $M_{\text{compact}}$, Count of Pruned Units $N_{\text{pruned}}$

 1: **for** each layer $l$ in $M$ **do**
 2:   **if** $l$ is not the output layer **then**
 3:     **for** each structure $s$ in layer $l$ **do**
 4:       Calculate $A_s = \sum_{i \in S} P_{ii} \cdot \theta_i^2$
 5:     **end for**
 6:     Sort structures in $l$ by $A_s$ in ascending order
 7:     Determine the number of structures to prune based on $T$
 8:     Prune determined number of structures with the lowest $A_s$ values
 9:   **end if**
10: **end for**
11: Update $N_{\text{pruned}}$ with the count of pruned structures
12: Fine-tune the pruned model $M$
13: Initialize $M_{\text{compact}}$ with dimensions aligned to the unpruned structures of $M$
14: Transfer weights from unpruned structures of $M$ to $M_{\text{compact}}$
15: Save $M_{\text{compact}}$ with updated parameters

---

nuclei in the images. It is a classic binary classification dataset used extensively in breast cancer research [57].

- *MNIST*: A foundational benchmark dataset in machine learning, MNIST consists of 60,000 training and 10,000 test images of handwritten digits (0 to 9) in 28x28 pixel grayscale format [58].

- *FashionMNIST*: A drop-in replacement for MNIST, Fashion-MNIST offers a greater challenge with its 60,000 training and 10,000 test images in grayscale (28x28 pixels). Each image represents one of ten clothing categories [59].

- *CIFAR-10*: This dataset contains 60,000 color images (32x32 pixels) divided equally among 10 classes (e.g., airplane, bird, cat) [60]. For our ResNet experiments, we augment CIFAR-10 with random flipping and cropping.

- *CIFAR-100*: A more fine-grained version of CIFAR-10, this dataset includes 60,000 color images (32x32 pixels) across 100 classes, with 600 images per class [60]. We apply random flipping and cropping for augmentation.

- *IMDB Movie Review*: This dataset is a collection of 50,000 movie reviews, balanced between positive and negative sentiments. It is commonly used for binary sentiment classification tasks [61].

### D.2 Models

- *FCN for MNIST (784, 256, 10)*: This Fully Connected Network (FCN) is specifically designed for the MNIST dataset. It comprises an input layer with 784 nodes, a hidden layer with 256 nodes, and an output layer with 10 nodes, making it a 2-layer FC network. Its architecture is optimized to handle the simplicity and characteristics of handwritten digit images.

- *FCN for CANCER (30, 100, 2)*: Customized for the CANCER dataset, this FCN includes an input layer of 30 nodes, two hidden layers, each containing 100 nodes, and a final output layer of 2 nodes. The 3-layer structure of this network is instrumental in distinguishing between benign and malignant tumors based on cellular features.

- *LeNet*: As a foundational Convolutional Neural Network (CNN), LeNet has shown exceptional performance in digit and image recognition tasks. We have applied LeNet to the MNIST, Fashion MNIST, and CIFAR-10 datasets, leveraging its capability to handle varying complexities of image data [58]. LeNet on CIFAR-10 is not a very common benchmark for pruning; here, it is used to demonstrate how SpaM, and specifically SpaM-OPD, is able to prune at high percentages without a performance loss up to 80% in a model that struggles with representing the data's complexity, showing that our work extends beyond over-parametrized networks for the task at hand.

- *MLPMixer*: The MLPMixer serves as a streamlined alternative to more complex models like CNNs and transformers. It relies solely on Multi-Layer Perceptrons (MLPs) for integrating inputs across spatial and channel dimensions [62]. We implement an MLPMixer with 2 blocks designed for MNIST.

- *ResNet with inplanes 64 and depth 18 for CIFAR-10*: We modify the implementation of ResNet and incorporate fixup initialization and custom bias and scale parameters to align with the constraints of the *ASDL* backend [63] used for the Laplace computations in this work, which does not support batch normalization.

- *Wide ResNet*: decreases depth compared to ResNet and increases the width of residual networks [64] with a depth of 16 and a widening factor of 4 (WRN16-4). We use fixup blocks to be able to utilize *ASDL* backend [63].

- *Vision Transformer* (ViT) [65]: unlike CNNs, which extract local features through filters and pooling layers, ViT breaks down images into fixed-size patches, treating each as a "token" in a sequence [65]. This allows it to leverage the Transformer architecture, initially designed for language processing, to analyze relationships between patches through *self-attention* mechanisms [66].

- *DistilBERT*: DistilBERT [67] is a smaller, faster, and cheaper version of BERT, achieved by leveraging knowledge distillation during the pre-training phase. This model retains 97% of BERT's language understanding capabilities while being 60% faster and 40% smaller. We use the pre-trained DistilBERT hosted in Hugging Face under (`distilbert-base-uncased`) [67] and tune it for sentiment analysis to classify reviews in the IMDB dataset [61] , which involves predicting the sentiment (positive or negative) of user reviews based on their textual content.

- *GPT-2*: a large-scale transformer-based language model developed by OpenAI, with impressive text generation capabilities. Trained on a vast corpus of internet text [68]. In our study, we leverage the 124M parameter version of GPT-2, fine-tuning it on the IMDB dataset for sentiment analysis to assess its performance under different pruning conditions.

### D.3 Dependencies

For the computation of second-order information (e.g., Hessian, Fisher information) needed for the Laplace approximation, we utilize the ASDL Library [63]. We use the library in its version under https://github.com/kazukiosawa/asdl/tree/011a942b2698b9ec33b0c8c47c96bd49335e5d80. The ASDL Library is distributed under the MIT License, which allows for reuse with a few restrictions that we respect in our work.

### D.4 Hyperparameters

**Marginal Likelihood**

- Hessian Approximation: The choice between GGN and EF. GGN was initially employed for fully connected networks, LeNets, and ResNets. However, for complex architectures (WRNs, ViTs, DistilBERT), GGN's computational cost became prohibitive, exceeding MAP runtime by up to 20x and even more for casual modeling tasks. Switching to EF maintained pruning performance while closely matching MAP runtime, which is particularly beneficial as GGN scales linearly with the number of classes. We discuss further the cost in Appendix D.7.

- *n_epochs_burnin* Dictates the number of epochs after which marginal likelihood optimization starts. If set superior to the number of training epochs, marginal likelihood is skipped, and the training is equivalent to MAP.

- *marglik_frequency* Controls the frequency of marginal likelihood estimation. The default value of 1 signifies re-estimation after each epoch, while a value of 5 indicates approximation for every fifth epoch.

We use these parameters to manage the computational cost of our experiments, where for small models like LeNets, FC Networks, the *n_epochs_burnin* is set to zero and *marglik_frequency* to one reflecting estimating each epoch since the start of the training. In contrast, for complex networks like

MLPMixer, ResNets, WideResNet, and ViT that we train from scratch, we start after 20 epochs and at an interval frequency of 5 epochs.

**Hyperparameters** Table D4 presents the specific hyperparameters employed for each dataset-architecture combination. We use † to denote the use of data augmentation in the training process. The symbols ⋆ and ⋄ represent the use of the Generalized Gauss-Newton (GGN) and Empirical Fisher (EF) approximations for the Hessian, respectively. We use *cosine decay* scheduler towards a fixed *minimum learning rate* of 1e-6 across all experiments. The symbols $\mathcal{D}_1$, $\mathcal{D}_2$, etc., represent the following datasets:

- $\mathcal{D}_1$: Breast Cancer Wisconsin (Diagnostic)
- $\mathcal{D}_2$: MNIST
- $\mathcal{D}_3$: FashionMNIST
- $\mathcal{D}_4$: CIFAR-10
- $\mathcal{D}_5$: CIFAR-100
- $\mathcal{D}_6$: IMDB Movie Review

All models are trained from scratch, denoted by the symbol ▲, except for DistilBERT and GPT-2, which are fine-tuned from pre-trained weights and are indicated by ▼.

Table D4: Hyperparameters used in the experiments.

| Dataset (Arch.) | Marglik Freq. | Batch Size | Learning Rate | Optimizer | Temp. | Burn-in / Epochs |
|---|---|---|---|---|---|---|
| $\mathcal{D}_1^{▲}$ (FCN) | 1 ⋆ | 64 | 0.001 | Adam | 1.0 | 0 / 50 |
| $\mathcal{D}_2^{▲}$ (FCN) | 1 ⋆ | 64 | 0.001 | Adam | 1.0 | 0 / 100 |
| $\mathcal{D}_2^{▲}$ (LeNet) | 1 ⋆ | 128 | 0.001 | SGD | 1.0 | 0 / 100 |
| $\mathcal{D}_3^{▲}$ (LeNet) | 1 ⋆ | 128 | 0.001 | SGD | 1.0 | 0 / 100 |
| $\mathcal{D}_3^{▲}$ (MLPMixer) | 1 ⋆ | 128 | 0.001 | Adam | 1.0 | 0 / 100 |
| $\mathcal{D}_4^{†}$ (ResNet) | 5 ⋆ | 128 | 0.1 | SGD | 5 | 20 / 100 |
| $\mathcal{D}_5^{†}$ (WRN) | 5 ⋄ | 128 | 0.1 | SGD | 5 | 20 / 200 |
| $\mathcal{D}_2^{▲}$ (ViT) | 5 ⋄ | 128 | 0.001 | Adam | 1.0 | 20 / 100 |
| $\mathcal{D}_6^{▼}$ (DistilBERT) | 5 ⋄ | 32 | 2e-5 | AdamW | 1.0 | 5 / 20 |
| $\mathcal{D}_6^{▼}$ (GPT-2) | 5 ⋄ | 8 | 2e-5 | Adam | 1.0 | 5 / 10 |

**Computational resources** Our experiments are run on GPUs. We run our experiments in a single GPU configuration on available variation between 1080 Tis, V100s, and A100s, with the majority being run on A100s with 40GB memory as we run the experiments intensively one after the other for different architecture on the same allocated GPU and in order to provide enough GPU memory. For models such as FCs, LeNets, ResNets, and MLP-Mixer, a GPU with 12GB of memory ( 1080 Ti) proved sufficient to run our method for our recommended laplace and prior, which is diagonal with parameter-wise priors and reproduce the results. For the sentiment analysis task using GPT-2, we recommend using a 32 GB GPU for tuning to be able to utilize a high batch size and to use diagonal approximation to fit laplace on the data without running into memory shortage.

**Runtime**

Table D5 presents the training and pruning runtimes on A100 for each dataset-architecture combination. Training times are given for both SpaM diagonal with parameter-wise prior and MAP, while pruning time is identical to both. Pruning runtimes refer to the time taken for OPD to compute and prune a model at 10 target sparsities. OPD and magnitude are very close in terms of runtime and the most efficient compared to SNIP, which is slightly slower due to it requiring an additional forward pass, and GraSP, which is significantly slower as it accumulates the gradient as shown in Figure D13.

### D.5 Pruning Criteria

- **SNIP**: Uses connection sensitivity, *how much a specific weight contributes to the output loss*, for effective pruning [36].
- **GraSP**: Employs gradient signal preservation. GraSP relates to the concept of Gradient Flow (GF), defined as:

$$GF = gL(\Theta)^T gL(\Theta) = ||gL(\Theta)||_2^2, \tag{D1}$$

Table D5: Runtimes for the experiments. Train: Training time (h:m), Prune: Pruning time (m:s).

| Dataset (Arch.) | Train | | Prune |
|---|---|---|---|
| | SpaM | MAP | OPD |
| $\mathcal{D}_1^{\blacktriangle}$ (FCN) | 0:01 | 0:01 | 0:04 |
| $\mathcal{D}_2^{\blacktriangle}$ (FCN) | 0:15 | 0:5 | 0:23 |
| $\mathcal{D}_2^{\blacktriangle}$ (LeNet) | 0:16 | 0:10 | 0:15 |
| $\mathcal{D}_3^{\blacktriangle}$ (LeNet) | 0:16 | 0:10 | 0:21 |
| $\mathcal{D}_3^{\blacktriangle}$ (MLPMixer) | 0:05 | 0:07 | 0:10 |
| $\mathcal{D}_4^{\dagger}$ (ResNet) | 1:24 | 0:25 | 0:55 |
| $\mathcal{D}_5^{\dagger}$ (WRN) | 1:17 | 1:12 | 0:51 |
| $\mathcal{D}_2^{\blacktriangle}$ (ViT) | 0:26 | 0:15 | 0:24 |
| $\mathcal{D}_6^{\blacktriangledown}$ (DistilBERT) | 5:20 | 2:15 | 1:05 |
| $\mathcal{D}_6^{\blacktriangledown}$ (GPT-2) | 17:34 | 6:24 | 17:41 |

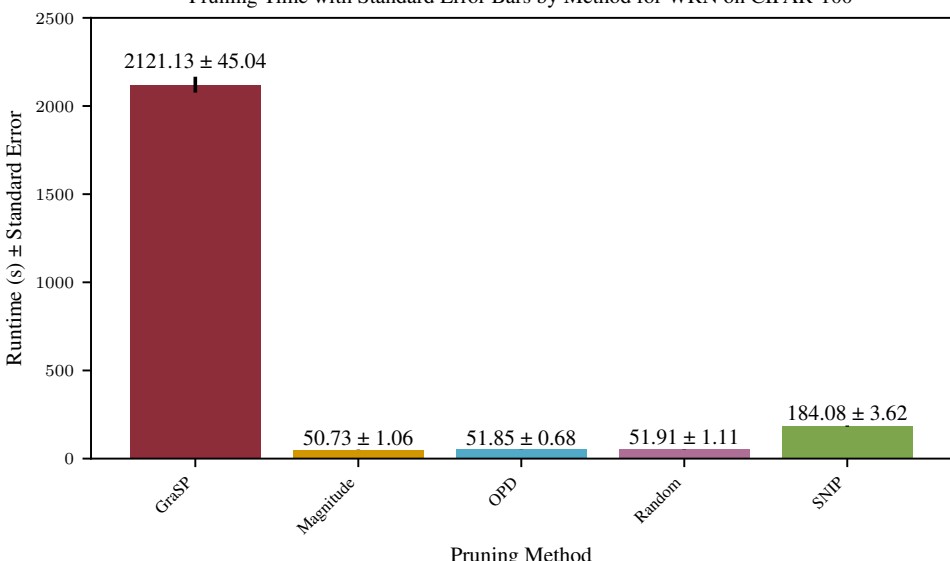

Figure D13: Mean relative pruning time with standard error bars on WRN with CIFAR-100. OPD and Magnitude are the most efficient as they use pre-computed parameters, with SNIP being slower due to requiring an additional forward pass, while GraSP is significantly slower as it needs to accumulate the gradient.

emphasizing the impact of pruning on the training dynamics [37]. We replicate the GraSP implementation of Rachwan et al. [39], where we consider the absolute value of the importance score initially proposed by Lubana and Dick [38] given by:

$$I(\Theta_t) = |\Theta_t^T H_L(\Theta_t) g_L(\Theta_t)| \tag{D2}$$

Note that while the importance score was initially used before training, we propose to use this importance score as a one-shot criterion after the training process and show how SpaM can leverage the performance of GraSP.

- **Structured-SynFLow**: We challenge the capabilities of SynFlow [50], a data-agnostic pruning approach that prevents layer collapse that happens at high sparsities where layers are no longer able to perform at the model's predictive power. This typically occurs when the pruning algorithm, intentionally or inadvertently, removes a significant portion of weights or filters from a specific layer, effectively collapsing its functionality [50]. We push SynFlow to its limits through advanced structured pruning strategies, where we prune layers aggressively

at the same target sparsity, which facilitates the compression process and resizing. By applying rigorous layer-specific filtering and neuron pruning, we aim to test the robustness and effectiveness of SynFlow in extreme sparsity *structured* scenarios. This approach not only benchmarks SynFlow's performance under stringent conditions but also explores its potential to maintain network functionality and accuracy in highly sparse neural network architectures.

- **Magnitude Pruning**: Relies on the magnitude of weights for pruning, aiming to maintain model performance while reducing complexity [69]. After the success shown by Han et al. [69], many methods adapted magnitude as a pruning criterion coupled with different scheduling [49, 70, 48].
- **Random Pruning**: Prune weights or structure randomly.

### D.6 Structured Sparsification Process

For structured sparsification, contrasting with the unstructured approach, the process necessitates reshaping the weight matrices to effectively reduce model complexity. The steps include:

1. One-shot structure masking based on aggregated importance scores.
2. Continue training for five epochs using the model from Step 1 for preliminary evaluation.
3. Implementing two software design approaches:
   - In-place layer replacement in the model with smaller ones fitting the non-masked regions.
   - Creating a new, flexible model initialized to match the dimensions of the non-masked areas, requiring repeated reading of the nonzero mask for state-dictionary and metadata alignment.
4. Transferring non-zero structures to smaller layers and tuning the model.

Post structure removal, we extend the training phase to adapt the model weights and re-evaluate, ensuring seamless functionality once transferred to smaller layers. Particularly after significant structural reduction, our primary objective shifts to maximizing performance in the downsized model. This fine-tuning spans 5 or 10 epochs depending on the complexity of the original model's structure, which was initially trained for either 50 or 100 epochs.

### D.7 Computational Cost

Instead of using the Generalized Gauss-Newton (GGN) approximation, which scales linearly with the number of classes, we can also use the Empirical Fisher (EF). For most architectures, using EF instead of GGN for SpaM does not add a very large computational overhead to MAP, as EF costs roughly as much as gradient computation. This is particularly beneficial as GGN scales linearly with the number of classes. The pruning results are not significantly affected by the choice of GGN or EF.

The runtime of MAP and SpaM was close (roughly 1h and 20 minutes on A100s) for WRN-16 on CIFAR100 using SpaM (EF) with diagonal LA and parameter-wise priors (our recommended settings for pruning). For language transformers, specifically DistilBERT and GPT-2, SpaM with EF does result in a longer training time compared to MAP. However, this increase is considerably less than when using GGN, where a single epoch can take longer than the entire SpaM training with EF.

In prior works [71], it was found that GGN gives a better posterior predictive approximation, but we do not use it in this work. We find that EF works similarly well for pruning at a much lower cost.

