# OpenReview forum: "Shaving Weights with Occam's Razor: Bayesian Sparsification for Neural Networks using the Marginal Likelihood"
_NeurIPS.cc/2024/Conference — NeurIPS 2024 poster_

### Official Review · Reviewer_bZUn · 2024-06-24

**Soundness:** 4
**Presentation:** 4
**Contribution:** 4
**Rating:** 8
**Confidence:** 4

**Summary:**

This paper proposes to train neural networks that are more amenable to pruning, using a Bayesian approach. In particular, they specify separate priors over individual parameters rather than a shared global prior, which allows some parameters to be regularised more than others. They maximise a Laplace approximation to the marginal likelihood to optimise these priors, deriving an approximation that allows them to utilise richer block-diagonal KFAC precision matrices rather than simple diagonal ones. Furthermore, they propose a scoring criterion based on the posterior variance and magnitude of the weights for deciding which weights to prune. The authors found that their method for pretraining sparsifiable networks generally led to much better performance after pruning than traditional MAP training.

**Strengths:**

- The proposed "SpaM" algorithm for pretraining sparsifiable networks in conjunction with their "OPD" criterion appears to be an effective method for pruning a variety of network types, based on the empirical results in the paper.
- The experiments are extensive and relevant to the contributions of the paper
- The approximation that allows KFAC to be used with non-scalar priors is useful to the KFAC literature in general, not just in the scope of network pruning.
- The paper is well written and concepts are introduced in an intuitive fashion.

**Weaknesses:**

- The more complicated KFAC with non-scalar prior approximation does not appear to be empirically better than using a diagonal precision matrix in the Laplace approximation, where the use of non-scalar priors is much simpler. However, it is still an interesting and potentially useful approximation outside of this particular application.

### Nitpicks

- Some details may be unclear to people less familiar with the area, such as the "interleaved" training of the prior hyperparameters and network parameters, or what exactly is being optimised in the MAP comparisons.
- Some of the more interesting figures (in my opinion) are relegated to the appendix, such as the fine-tuning figure, whilst the main paper has elements that feel superfluous or irrelevant, such as Figure 1, which seems unnecessary to explain the method, or a rather lengthy discussion of the use of marginal-likelihood in e.g. section 3.2.

**Questions:**

### Questions
- Do the authors believe that the disappointing performance of the KFAC with non-scalar priors was due to the extra approximation that was necessarily made (Proposition 3.1)?
- In the MAP experiments, are we using the same Laplace approximation that was used to approximate the marginal log-likelihood, and if so, how are the prior hyperparameters chosen? Please clarify if I appear to be misunderstanding this aspect.
- In lines 165-166, could you expand upon what is meant by "optimize the Laplace approximation to the marginal likelihood after an initial burn-in phase with a certain frequency"?
- Does SpaM allow for improving structured sparsifiability **during training** as well as just unstructured? This wasn't entirely clear to me


### Suggestions
- Expand slightly more on the difference between uniform and global pruning.
- Remove some of the less relevant figures / discussion and include more of the experimental plots in the main text.
- Typo in Eq. 5. $\mathbf Q^T$ should be $\mathbf Q^T_A$?
- In Eq. 5, it would be clear to write $(\boldsymbol \Lambda_A \otimes \boldsymbol \Lambda_G + \delta \mathbf I)$ rather than $(\boldsymbol \Lambda_A \otimes \boldsymbol \Lambda_G + \delta)$ as the latter implies elementwise addition of $\delta$ rather than just the diagonal. Please let me know if this is a misunderstanding on my part.
- Many of the figure references in the paper didn't seem to lead to the right place, e.g. on line 239. This might just be a problem with my machine, please double check.

**Limitations:**

The authors briefly discussed some generic limitations of Laplace approximations and additional computational cost of their SpaM training procedure over MAP. It would have been nice to comment on, for example, the remaining difficulty of structured pruning, since it is so much more beneficial than unstructured pruning.

---

> ### Author Rebuttal · Authors · 2024-08-07
>
> We thank you very much for your detailed and helpful feedback. We really appreciate your suggestions and will incorporate them in the revision.
>
> **Weaknesses:**
> > The more complicated KFAC with non-scalar prior approximation does not appear to be empirically [...] However, it is still an interesting and potentially useful approximation outside of this particular application.
>
> Thank you for your comment and raising this point. You understood our motivation with this proposal. We would like to expand further by adressing your question below.
>
> **Nitpicks**
> > Some details [...] "interleaved" training of the prior hyperparameters and network parameters, or what exactly is being optimised in the MAP comparisons.
>
> The optimization is conducted after a burn-in phase (number of initial epochs) and at a specific frequency (after each x epochs) referred to as `marglik_frequency` in the hyperparameter sections.
> In is worth mentioning that after pruning (at test time), we just use the trained posterior mean (MAP solution under the optimized prior) and not the full Laplace predictive.
>
> > Some of the more interesting figures (in my opinion) are relegated to the appendix, [...] such as Figure 1, which seems unnecessary to explain the method, or a rather lengthy discussion of the use of marginal-likelihood in e.g. section 3.2.
>
> We thank you for the suggestions, it was challenging to fit all key results in the main body of the paper.
> We will utilize the space better in the revised version.
>
> **Questions:**
> >  Do the authors believe that the disappointing performance of the KFAC with non-scalar priors was due to the extra approximation that was necessarily made (Proposition 3.1)?
>
> Proposition 3.1 came into place to bridge the gap and to emphasize the contribution of diagonal priors in making neural networks more prunable, as it was shown to perform exceptionally well when combining the diagonal approximation with a diagonal prior. It was shown that a diagonal prior in KFAC pushes the probability of the network compared to other priors like scalar and layerwise (the approximation results are referred to as parameter-wise in red in Figure B4, compared to the KFAC with solid lines).
> For our work, we do recommend the usage of the diagonal approximation with diagonal priors as we do not need the precision of KFAC compared to diagonal in offering a better approximation. However, other works, as you mentioned, might benefit from it all while pushing the probability of networks.
>
> > In the MAP experiments, are we using the same Laplace approximation that [...] how are the prior hyperparameters chosen? Please clarify if I appear to be misunderstanding this aspect.
>
> For MAP, we use a standard training approach where we do not optimize the marginal log-likelihood, and thus generally do not use Laplace. In the case of using OPD with MAP, the inverse of the Hessian is computed to be able to perform Laplace post-hoc on a pre-trained network without the need for the network to be trained and optimized for the marginal likelihood.
>
>
> >[...] "optimize the Laplace approximation to the marginal likelihood after an initial burn-in phase with a certain frequency"?
>
> Thank you for your insightful question. We will include an explanation in the revision or refer to appendix D.4. The approximation is conducted during training after an initial number of epochs (burn-in phase) and at a specific frequency (e.g., every 5 epochs) for computational considerations. For smaller architectures, the approximation can be used from the start of training and at each epoch, but for larger architectures, this would be challenging and require more ressources.
> The table D.4 shows the choice of the parameters for each experiment and explains their significance.
>
>
> > Does SpaM allow for improving structured sparsifiability during training as well as just unstructured? This wasn't entirely clear to me.
>
> SpaM can be applied during training for the structured case, but it is more efficient to mask structures (zero them out) instead of compressing them (removing them) during training. Compression during training is inefficient due to the need to replace layers and copy weights. By masking structures, we can evaluate the smaller network before fully committing to removing any components, and this does not add much overhead, as we have everything pre-computed with each marginal likelihood update and only need to aggregate the score. After training, compression can be performed once.
>
>
> >  Expand slightly more on the difference between uniform and global pruning.
>
> Thanks, we will expand on this in the revised version.
>
> **Global pruning** prunes across the entire network at a target percentage (e.g., 80%). This approach may result in some layers being pruned more heavily than others to achieve the overall target sparsity.
>
> **Uniform pruning**, on the other hand, applies the same pruning percentage to each layer.
>
> > Remove some of the less relevant figures/discussion and include more of the experimental plots in the main text.
>
> Thank you, we will adress your suggestion in the revised version.
>
> > Typo in Eq. 5. should be $\mathbf{Q}^T_A$?
>
> Thank you for pointing this out, that's correct and we will add the subscript $A$.
>
> >  In in Eq. 5, it would be clear to write [...] Please let me know if this is a misunderstanding on my part.
>
> Thank you for the suggestion. We will add the identity for clarity.
>
> >  Many of the figure references in the paper didn't seem to lead to the right place, [...] please double check.
>
> Thank you. Indeed, some links to the appendix figures seem to be broken, where the reference is rendered correctly, but the hyperlinks seem to be broken due to a counter change. We solved this issue by setting `hypertextnames` to `false` in the `hyperref` package.

---

> > ### Comment · Reviewer_bZUn · 2024-08-09
> >
> > Thank you for your detailed rebuttal. You answered many of my questions satisfactorily.
> >
> > > Does SpaM allow for improving structured sparsifiability during training as well as just unstructured?
> >
> > Sorry, my question doesn't seem to have been clear. I meant "does SpaM do anything to encourage weights to be amenable to structured pruning in particular, or does the question of "structured or unstructured" only affect the final pruning using OPD?". I believe I'm specifically talking about when you pruning post-hoc, rather than online.
> >
> > Thinking more about Figure 1, I personally think it would be useful to have a diagram entirely devoted to the SpaM training process (e.g. compute laplace approximation, do backprop for K steps, update laplace approximation etc.), rather than focusing on the actual pruning using OPD after, as the main idea of the paper to me seems to be encouraging weights to be sparsifiable during training, using clever priors.

---

> > > ### Author Response · Authors · 2024-08-09
> > >
> > > Thanks for the follow-up question.
> > >
> > > > Does SpaM do anything to encourage weights to be amenable to structured pruning in particular, or does the question of "structured or unstructured" only affect the final pruning using OPD?
> > >
> > > Apart from differently aggregated weights in OPD between structured and unstructured, SpaM can also encourage structured pruning by specifying priors that correspond to sensible groups, for example, rows or columns of weight matrices. One such prior would be the unit-wise prior defined in lines 130-134. One unit can correspond to a neuron in the fully-connected case or a filter in the convolutional case
> > >
> > > > I personally think it would be useful to have a diagram entirely devoted to the SpaM training process.
> > >
> > > Thanks for the suggestion. We will try to include this process visually or make a separate figure or algorithmic description for it in the next revision.

---

> > > > ### Comment · Reviewer_bZUn · 2024-08-09
> > > >
> > > > Thank you for the clarification.
> > > >
> > > > I maintain my score of 8 (strong accept) which reflects my opinion that this is good work that has been carried out to an excellent standard.

---

### Official Review · Reviewer_NVyT · 2024-06-27

**Soundness:** 4
**Presentation:** 4
**Contribution:** 3
**Rating:** 6
**Confidence:** 4

**Summary:**

This paper proposes to sparsify a neural network using Bayesian principles by optimizing the marginal likelihood (SpaM). Specifically, the authors use weight/node/layer-wise Gaussian priors over the weights and learn the corresponding scales by maximizing the marginal likelihood during training using Laplace approximation. Compared with MAP, where a shared regularization is used, SpaM optimizes the prior scale for each weight/node/layer to regularize weights adaptively. Moreover, a new important score, Optimal Brain Damage (OBD), is proposed for pruning weights, by using the approximated posterior. The effectiveness of this method is demonstrated with extensive experiments on various datasets and model architectures.

**Strengths:**

The paper is novel and well-written.

Although sparsity-inducing priors have been widely used to sparsify deep neural networks, optimizing the hyper-parameters in the prior using ML-II with Laplace approximation is novel and seems to be promising compared with using a single fixed scale to shave weights.

The experiments are comprehensive, including results with different prior structures, pruning criteria, architectures (conv and transformer), and dataset domains (images and texts).

**Weaknesses:**

This paper is generally well written with extensive experiment, and I only have one following concern (I'm very happy to increase my score if this is addressed):

Hyper-parameters in the prior often can be tackled in two ways as a Bayesian: 1. optimize the hyper-parameters with ML-II; 2. conduct Bayesian inference on the hyper-parameters by having hyper-priors on them.

This paper focuses on the first approach, but the comparison with the second approach is missing. In fact, the second approach has been widely used [1-4] to sparsify deep neural nets; for example, the node/layer-wise horseshoe prior and spike-and-slab prior can also offer different regularization strengths on different weights as well as give a more structured sparsity with good theoretical guarantees. Moreover, ML-II is known to have the risk of overfitting compared with the full Bayesian approach. So, I believe it is important to compare ML-II with the full Bayesian inference.

[1] Ghosh, Soumya, Jiayu Yao, and Finale Doshi-Velez. "Model selection in Bayesian neural networks via horseshoe priors." Journal of Machine Learning Research 20.182 (2019): 1-46.

[2] Louizos, Christos, Karen Ullrich, and Max Welling. "Bayesian compression for deep learning." Advances in neural information processing systems 30 (2017).

[3] Cui, Tianyu, et al. "Informative Bayesian neural network priors for weak signals." Bayesian Analysis 17.4 (2022): 1121-1151.

[4] Polson, Nicholas G., and Veronika Ročková. "Posterior concentration for sparse deep learning." Advances in Neural Information Processing Systems 31 (2018).

**Questions:**

Is optimizing scales with ML-II using Laplace approximation better than doing a full (approximate) Bayesian inference over scales with common sparsity-inducing priors (e.g., horseshoe) using VI or SGHMC, in terms of computation, accuracy/uncertainty given a sparsity level, etc.?

**Limitations:**

The authors mentioned the limitation coming from the Laplace approximation.

---

> ### Author Rebuttal · Authors · 2024-08-07
>
> Thank you for your evaluation of our work and your constructive comment that shows a good understanding of the field and your readiness to increase the score if your concerns and questions are adressed. We would like to further explain our motivation behind using Laplace approximations.
>
> **Weaknesses**:
>
> > Hyper-parameters in the prior often can be tackled in two ways as a Bayesian: 1. optimize the hyper-parameters with ML-II; 2. conduct Bayesian inference on the hyper-parameters by having hyper-priors on them.
> [...] ML-II is known to have the risk of overfitting compared with the full Bayesian approach. So, I believe it is important to compare ML-II with the full Bayesian inference.
>
> Thank you for your comment. The main reason not to go for the full Bayesian inference approaches is their known computational complexity and lack of scalability for large-scale problems [1] which is emphasized in the suggested work (Ghosh et al., 2019; Louizos et al., 2017; Cui et al., 2022; Polson & Ročková, 2018). With a full Bayesian approach, it is not possible to provide a solution that can be scaled across the different architectures considered in our work (e.g., ViT and GPT-2). Utilizing ASDL [2] to compute the second-order information needed for the Laplace approximation, we make use of modern scalable algebra engines and avoid the repeated sampling and complex variational optimization that tends to slow down training and reduce performance.
>
> This makes our method easier to use for practitioners, since it scales better with large datasets and complex models and supports different types of layers. For instance, it made it possible for us to use our method starting from a "pre-trained" backbone (e.g., from huggingface, without adjustments to the architecture) and transfer learn or fine-tune on another task.
>
>
>
>
> **Questions:**
> > Is optimizing scales with ML-II using Laplace approximation better than doing a full (approximate) Bayesian inference over scales with common sparsity-inducing priors (e.g., horseshoe) using VI or SGHMC, in terms of computation, accuracy/uncertainty given a sparsity level, etc.?
>
> Yes, full (approximate) Bayesian inference methods are likely intractable for large-scale problems due to their computational demands and the complexity of approximating or sampling from high-dimensional posteriors, as laid out in our response above.
>
> In our case, we wanted the method to be able to scale and not be restricted by the limitations that come with full Bayesian inference that would require downscaling the experiments and not being able to perform our approach on modern architectures like vision and language transformers. BNNs are usually challenging to implement and expensive to train. With the Laplace approximation, it is much easier to use BNNs [3] and, during inference, to use a single forward pass on the pruned architecture.
> This approach allowed us to even be able to perform OPD on pre-trained models (by computing the inverse Hessian needed for Laplace post-hoc) that are very expensive to train, like GPT-2, and as well to use SpaM on architectures like DistilBERT and ViT.
>
> **References**
>
> [1] Bai, J., Song, Q., & Cheng, G. (2020). Efficient variational inference for sparse deep learning with theoretical guarantee. Advances in Neural Information Processing Systems, 33, 466-476.
>
> [2] Osawa, K., Ishikawa, S., Yokota, R., Li, S., & Hoefler, T. (2023). ASDL: A Unified Interface for Gradient Preconditioning in PyTorch.
>
> [3] Daxberger, E., Kristiadi, A., Immer, A., Eschenhagen, R., Bauer, M., & Hennig, P. (2021). Laplace Redux – Effortless Bayesian Deep Learning. NeurIPS.

---

> > ### Comment · Reviewer_NVyT · 2024-08-09
> > **Official comment**
> >
> > Thank you for your rebuttal, which answered my questions from a motivation perspective. I have increased my score accordingly.

---

### Official Review · Reviewer_oKfx · 2024-07-08

**Soundness:** 2
**Presentation:** 3
**Contribution:** 2
**Rating:** 5
**Confidence:** 4

**Summary:**

The paper introduces a new pruning technique named SpaM, which leverages Laplace approximation and Bayesian marginal likelihood for approximating the posterior in Bayesian Neural Networks. This approach includes two main components: the Gaussian prior variance and OPD, a pruning method that utilizes the Laplace posterior precision.

**Strengths:**

Strengths:
- Clarity and Accessibility: The paper is well-written, making it accessible to readers.
- Motivation and Methodology: The use of marginal likelihood for pruning neural network models is well-motivated. The proposed method is applicable to both unstructured and structured pruning scenarios.
- Bayesian Justification: The method is grounded in Bayesian theory, presenting a new pruning criterion based on posterior approximation.
- Simplicity: The approach is relatively simple compared to existing Bayesian pruning methods.

**Weaknesses:**

Weaknesses:

Methodological Clarity:

- The computational and storage challenges associated with the Fisher matrix or generalized Gaussian Newton matrix in the context of Laplace approximation for Bayesian Neural Networks are not sufficiently addressed in the limitations section.
- The integration of the method with existing pruning criteria is unclear. The process involving MAP solution, Laplace approximation, and OPD with the precision matrix needs clarification, especially in comparison with zero-shot pruning methods like GraSP, SNIP, and Random.
- The fairness of comparing the proposed method with Monte Carlo sampling for BMA performance computation against MAP is questionable. It should be clarified if only a single pruned model or Laplace approximation is used.

Experimental Design:

- The computational and memory costs of Laplace approximation should be compared with other baselines.
- The baseline methods used are too simplistic. Stronger baselines such as IMP, RigL, or DST should be included.
- The experiments are limited to CIFAR10. Additional datasets like CIFAR100 and ImageNet would strengthen the findings.
- The datasets and networks used are relatively simple, raising questions about the method's performance on more complex datasets and models.
- Specific questions about the online pruning process need addressing, such as the timing of pruning, the validity of Laplace approximation during training, and optimization of prior parameters.

Novelty and Comparison:

- The novelty is somewhat limited, as the method relies on known ideas from the literature.
- The differences in uncertainty estimation between SpaM and MAP are marginal. The presentation of results could be improved to better highlight these differences.
- Previous studies using Bayesian inference or variational methods for pruning should be compared with SpaM in both related work and experiment sections to enhance understanding.

The scale of Experiments:

- The experiments are considered limited in scale, with MNIST and CIFAR being seen as insufficient to provide meaningful insights. Larger models and datasets should be used for validation.
Performance Goal:

- The ultimate objective in neural network pruning is to achieve performance comparable to the unpruned dense model at a given sparsity level. The paper should position SpaM+OPD in relation to other pruning methodologies more clearly.

**Questions:**

Questions:

- Previous Studies Comparison: A comprehensive comparison between SpaM and previous studies using Bayesian inference or variational methods for pruning would be beneficial.
- Clarification of Procedures: The paper should explicitly explain the sequence of dense training followed by prune-after-training, as this is critical for reader understanding.

**Limitations:**

The authors have sufficiently addressed the potential negative societal impacts of their work. For details on the limitations, refer to the Weaknesses and Questions sections.

---

> ### Author Rebuttal · Authors · 2024-08-07
>
> **Methodological Clarity:**
>
> >  Computation/Storage challenges
>
> We discuss computational and storage costs of SpaM in depth in the main text but also in appendices D.4 & D.7. A diagonal approximation costs as much to store as a parameter vector and KFAC costs roughly twice as much so we incur minimal storage overhead. Computationally, using the EF is as cheap as gradient computation while the GGN scales with the number of outputs. For the benefit provided, these are minor additional costs.
> >  Integration of the method is unclear.
>
> There seems to be a misunderstanding as we have already covered these details. In appendix B.1, we explain that we initially train the NN either with MAP (a) or SpaM (b). Once these networks are trained, we prune both models using the different pruning criteria at different sparsity levels. Thus, we can measure the effect of SpaM training and OPD separately and fairly.
> >  Fairness of comparison  [...].
>
> We do not use the predictive posterior for inference or Bayesian model averaging (BMA) but simply use the point estimate network. During inference, the model trained with SpaM only uses a single forward pass over the pruned architecture, thus guaranteeing a fair comparison with the baselines. The posterior is solely used to estimate the marginal likelihood during SpaM training.
>
> **Experimental Design:**
> >  Computational and memory costs of Laplace approximation.
>
> We compared the cost of Laplace approximation used in SpaM (both using GGN and EF) with MAP and, as theoretically validated, it only adds minor overhead, especially in the diagonal EF variant.
> >  Baseline methods are too simplistic.
>
> We appreciate your feedback. To address this point and avoid any misunderstanding, we have addressed your comment in a detailed explanation in our general response.
> > Experiments are limited to CIFAR10. Additional datasets like CIFAR100 and ImageNet [...].
>
> The experiments are not limited to CIFAR10. We have included CIFAR100 and IMDB as larger and different datasets, respectively.
> >  Datasets and networks used are relatively simple.
>
> We indeed cover in our paper a large variety of datasets and models, including MLP, CNN, MLP-Mixers, Transformers (Vision Transformers (ViT), Language Transformer (DistilBERT)) and show that our method can be applied to pre-trained models like GPT-2. The list of models and datasets are represented in the appendices D.1 and D.2, respectively.
> >  Specific questions about the online pruning process.[...].
>
> Appendix B.6 addresses these questions and explains how the pruning is conducted during training. Pruning occurs incrementally after a new computation of the marginal likelihood.
>
> **Novelty and Comparison:**
> > The novelty is somewhat limited, [...].
>
> We address the novelty of our approach with the general response.
> > Differences in uncertainty [...] are marginal. The presentation of results could be improved [...].
>
> We acknowledge that the difference in uncertainty estimation between SpaM and MAP could be more clearly visualized and will adjust the y-axis scale accordingly (due to random pruning having high values). The difference is indeed significant at high sparsities for criteria like OPD, GraSP, and magnitude (see Figure B8 and Table B3).
> >  Comparison with previous studies using Bayesian inference or variational methods for pruning..
>
> Variational inference for deep neural networks usually slows down training and reduces performance in comparison to the MAP. Further, as found by Blundell et al. (2015) in their seminal Bayes-by-Backprop paper, it does not work in conjunction with prior hyperparameter updates and thus cannot be applied in the automatic regularization setting we are interested in. We will include a more thorough discussion in the related work section.
>
> **The scale of Experiments:**
> >  The experiments are considered limited in scale,[...]. Larger models and datasets should be used for validation.
>
> We cover a large variety of models and datasets (vision and text):
>
> **Models**: fully connected, LeNets, MLPMixer, (Wide) ResNets, ViTs, DistilBERT, GPT-2
>
> **Datasets**: UCI breast cancer, MNIST, FashionMNIST, CIFAR10, CIFAR100, IMDB
> >  [...]. The paper should position SpaM+OPD in relation to other pruning methodologies more clearly.
>
> Based on Figure 2 which compares SpaM and MAP across different criteria,
> SpaM creates models that are inherently more sparsifiable.
> OPD (blue lines) consistently ranks among the top-performing criteria, maintaining near-unpruned performance even at 95% sparsity in vision tasks. GPT-2 results also highlight OPD's standalone effectiveness without SpaM.
>
> Taken together, SpaM and OPD offer a powerful combination for achieving the ultimate pruning goal: high performance at high sparsity.
>
>
> **Questions**
> >  Previous Studies Comparison: [...] SpaM and previous studies using Bayesian inference or variational methods for pruning [...].
>
> A similar comparison would down-scale our evaluation pipeline, as most VI methods and those relying on full (approximate) Bayesian inference are limited and likely intractable for large-scale models. Such approaches are typically constrained to smaller architectures like MLPs and CNNs [1-3].
>
> Methods like MCMC are impractical for large-scale models due to high computational demands and numerous samples needed to estimate posterior distributions accurately. The Laplace approximation used in SpaM avoids sampling, making it computationally feasible.
>
> [1] Molchanov et al. (2017). Variational Dropout Sparsifies Deep Neural Networks. ICML.
>
> [2] Zhao et al. (2019). Variational Convolutional Neural Network Pruning. CVPR.
>
> [3] Bai et al. (2020). Efficient Variational Inference for Sparse Deep Learning with Theoretical Guarantee. NeurIPS.
>
> > [...] The paper should explicitly explain the sequence of dense training followed by prune-after-training, [...].
>
> We provide a detailed description of the training approach in the main text and more in detail in appendices B.1 and D.

---

> > ### Comment · Reviewer_oKfx · 2024-08-12
> >
> > Thank authors for addressing my concerns in the author response. I have increased my score to 5 as a result of the clarification. However, I still have some reservations about the practicality of the proposed method.

---

### Official Review · Reviewer_rzPD · 2024-07-15

**Soundness:** 3
**Presentation:** 3
**Contribution:** 3
**Rating:** 7
**Confidence:** 3

**Summary:**

The authors present a framework for assessing the sparsifiability of a parametric model, a measure of how many parameters can be pruned without severely affecting the modelling performance. In essence, the authors suggest training a Bayesian neural network (BNN) using the marginal likelihood estimated via the Laplace approximation. The marginal likelihood's automatic Occam's Razor ability to identify good trade-offs between model complexity and data fit will encourage sparsity, and the trained network can then be subjected to a pruning criterion of choice.

While any pruning criterion can be used, the authors propose the Optimal Posterior Damage (OPD), which is computed as a cheap byproduct of their marginal likelihood approximation. This often outperforms more expensive approaches.

The authors demonstrate the effectiveness of their proposed approach through several experiments covering performance at different sparsity levels, online pruning, uncertainty estimation, the influence of prior choice, and structured sparsification. Generally, their proposed approach and pruning criterion outperforms baselines.

**Strengths:**

**Originality**
1. Using the marginal likelihood to encourage less complex networks appears original in the context of pruning, although a little incremental.
2. The proposed pruning criterion, Optimal Posterior Damage (OPD), and the structured priors for the KFAC approximation appear to be original contributions.

**Quality**
1. Marginal likelihood optimisation is a well-known technique which has many theoretically and intuitively pleasing properties.
2. The OPD pruning criterion makes intuitive sense and seems remarkably powerful.
3. The experimental evaluation is quite extensive and includes repetitions over multiple seeds (and the resulting uncertainties over the reported metrics). I find the evaluation of different choices for the structure of the prior is particularly interesting, and the recommendation for a default configuration is great.

**Clarity**
1. The paper is very well-written, and the plots are generally informative and easy to understand.

**Significance**
1. The marginal likelihood idea is simple yet works well in practice, which is great for the potential significance of the approach.
2. The proposed pruning criterion seems bound to become a strong baseline whenever the training approach is used.
3. The structured priors for the KFAC approximation are a minor contribution, but will likely be used outside of the pruning literature too, thus making them quite significant.

**Weaknesses:**

**Originality**
1. While using the marginal likelihood as the training objective in the context of network pruning seems original, it seems somewhat incremental.

**Quality**
1. The paper proposes a combination of two methods, training a Bayesian network using (an approximation to) the marginal likelihood and a pruning criterion, but only the choice of pruning criterion is evaluated experimentally, not the training scheme. Is training using the marginal likelihood better than a simple maximum likelihood optimisation with, say, L1 regularisation? The authors don't answer this important question.
2. While the authors test the effect of different prior structures (which are great and interesting experiments!), they do not test the effect of different prior distribution families. They seem to use a normal distribution for all experiments, which is surprising since the normal distribution doesn't induce sparsity and has been shown experimentally to be quite a poor choice for BNNs. Some obvious choices would be to test the horseshoe prior or the Indian buffet process prior, which can both encourage sparsity (e.g., Ghosh et al., 2018; Kessler et al., 2021). Other interesting choices would be the Laplace distribution, for which the MAP solution with a diagonal prior would correspond to L1 regularisation, Student's t and the spike-and-slab prior. Taken together with the remark above, it seems like the paper is missing half of the experimental analysis.
3. Experimentally, I would have liked to see results for the different pruning criteria in the settings that were used in their original publications (e.g., same training scheme). It is unclear if the paper's results are on par with the literature's.
4. Minor weakness: it would have been helpful to see the performance of the unpruned networks (i.e., 0% sparsity) to understand the penalty caused by the sparsification fully.

Comment: if the authors decide to include more experiments in a future revision, I think sections 1 through 3 could be significantly shortened without losing too much context.

**Clarity**
1. While the paper is overall very well-written, the terms in Eq. (4) lack definitions, and the appendix appears a little rough.
2. The font size in the plots could be a little larger.


**References**
- Ghosh et al. (2018). "Structured variational learning of Bayesian neural networks with horseshoe priors." ICML.
- Kessler et al. (2021). "Hierarchical Indian buffet neural networks for Bayesian continual learning." UAI.

**Questions:**

1. Additional experiments are probably infeasible for the rebuttal period, but did you already experiment with other prior distribution families? Why did you choose a normal distribution rather than a sparsity-inducing family?
2. The results for the online pruning approach in figure B6 confuse me a little. The online approach seems to generally perform better for higher sparsity levels for CIFAR-10, with LeNet on CIFAR-10 being particularly extreme (test accuracy around 50% for 20% sparsity compared to 65% for 80% sparsity ). Do you know what happens here?

**Limitations:**

The authors have adequately addressed the limitations of their approach.

---

> ### Author Rebuttal · Authors · 2024-08-07
>
> **Weaknesses**
> > While using the marginal likelihood as the training objective in the context of network pruning seems original, it seems somewhat incremental.
>
> Thank you for the chance to expand more on the novelty offered with our approach and scalability compared to established works. In fact, our framework addresses existing scalability challenges, making it practical for real-world scenarios, and enables automatic relevance determination in deep learning for the first time.
>
> **Quality**
> > The paper proposes a combination of two methods [...] Is training using the marginal [...] a simple maximum likelihood optimisation with, say, L1 regularisation? The authors don't answer this important question.
>
> Thank you for your insightful question. We consistently compare MAP and marginal likelihood training throughout the paper (solid and dashed lines in the plots). MAP with Gaussian priors (L2 regularization) is our primary baseline, as it corresponds to the widely used standard weight decay.
> We ran an experiment with L1 for the rebuttal (see provided pdf), which gave worse results.
>
> > Exploring other prior distributions like the horseshoe prior or Laplace distribution, which have been shown to induce sparsity and improve performance in Bayesian neural networks [...] the Laplace distribution, for which the MAP solution with a diagonal prior would correspond to L1 regularisation, Student's t and the spike-and-slab prior.
>
> Thank you for the opportunity to clarify. We address your concerns about prior choice in the general response. Alternative prior families often require expensive inference, making them infeasible for large-scale models. The Laplace approximation, which we employ for scalability reasons, requires differentiability, excluding many of your suggested priors. We instead use the idea of relevance determination, that is regularization of individual weights or weight groups. With the Laplace approximation, this is scalable to networks at any scale, thanks to modern Hessian approximations.
>
> > Experimentally, I would have liked to see results for the different pruning criteria in the settings that were[...]. It is unclear if the paper's results are on par with the literature's.
>
> Thank you for your valuable comments. While SNIP and GraSP are primarily used for pruning at initialization (PAI), their criteria based on weight contribution to loss and gradient preservation can be applied broadly for weight importance evaluation in various scheduling scenarios. However, using them as PAI often requires initial calibration passes for the network, especially at high sparsities, to achieve the desired pruning target. Additionally, compared to post-training methods, PAI can be less efficient, especially for large models that need retraining from scratch or rely on the availability of the original dataset (we have included PAI setup in our submitted code). In contrast, methods like SpaM can be more efficient for networks built on top of transfer learning, while OPD can be directly applied to pre-trained models by computing the inverse Hessian, as demonstrated in our GPT-2 experiments.
> Our main aim was to show the benefits of SpaM in a harmonized experimental setup, regardless of the specific pruning criterion.
>
>
> > Minor weakness: it would have been helpful to see the performance of the unpruned networks (i.e., 0% sparsity) to understand the penalty caused by the sparsification fully.
>
> Thank you. We found that most methods, especially OPD, perform at almost identical performance to the baseline at 20% and hence did not report 0% sparsity so far. We will add a horizontal line with the baseline accuracy that will also make it easier to visualize the performance gap at higher sparsities.
>
> **Clarity**
> > While the paper is overall very well-written, the terms in Eq. (4) lack definitions, and the appendix appears a little rough.
>
> Thank you  we will improve the definition in the revision and polish the appendix. $A_l$ and $G_l$ are the uncentered covariances of the layer inputs and output gradients, respectively. These are used as defined in the referenced works, for example [1].
>
> [1] Martens, James, and Roger Grosse. "Optimizing Neural Networks with Kronecker-factored Approximate Curvature." 2020
>
> > The font size in the plots could be larger.
>
> We will increase the font size to improve readability and accessibility.
>
> **Questions:**
> > Additional experiments are probably infeasible for the rebuttal period, [...] experiment with other prior distribution families? Why did you choose a normal distribution rather than a sparsity-inducing family?
>
> See our arguments regarding scalability above. Specifically, in the context of automatically learning regularization, the Gaussian group-wise prior, as in automatic relevance determination (ARD), has proven effective. Other priors, due to non-differentiability, could require intractable manual hyperparameter tuning.
>
> > The results for the online pruning approach in figure B6 confuse me a little. The online [...] for CIFAR-10, with LeNet on CIFAR-10 being particularly extreme [...]. Do you know what happens here?
>
> The x-axis in the online pruning curves (section B.6) indirectly reflects training progress, not just sparsity. LeNet on CIFAR-10 is pruned starting from epoch 10 for 100 epochs, aiming for 99% sparsity.  LeNet's curve trend on CIFAR-10 is therefore due to ongoing convergence during pruning, caused by dataset complexity relative to the architecture.

---

> > ### Comment · Reviewer_rzPD · 2024-08-08
> >
> > Dear authors,
> >
> > Thank you for your reply and for performing the additional experiment with L1 regularisation. Which of the plots in your original submission should I compare the rebuttal plot to? In particular, I'm looking for the MAP (L2 reg.) results for the same experiments.
> >
> > It's quite interesting that the L1 regularisation performs this badly, I think. Do you have an intuition for why this is? Perhaps L1 regularisation is simply too aggressive? To be clear, I'm very happy that your method works better, I'm just trying to understand the reason, if possible.
> >
> > In any case, I have increased my score. You have nicely addressed my questions and concerns, and I also think my comment that your contribution seems incremental was perhaps too harsh.

---

> > > ### Author Response · Authors · 2024-08-09
> > >
> > > Thank you for raising your score. We are happy that we have addressed your concerns.
> > > >Which of the plots in your original submission should I compare the rebuttal plot to? In particular, I'm looking for the MAP (L2 reg.) results for the same experiments.
> > >
> > > The experimental plots to compare to would be Figure 2 bottom left (WRN on CIFAR-100) and Figure B6 top right corner.
> > >
> > > >It's quite interesting that the L1 regularisation performs this badly, I think. Do you have an intuition for why this is? Perhaps L1 regularisation is simply too aggressive? To be clear, I'm very happy that your method works better, I'm just trying to understand the reason, if possible.
> > >
> > > We have thoroughly searched for an L1 regularization coefficient but could not find a setting that led to better performance. Indeed, L1 regularization seems to be too aggressive and leads to an overly pruned network in the end. One advantage of our method in comparison to L1 regularization is that it applies different regularization per weight and adapts to the data instead of one global fixed parameter. Theoreticaly, Wipf and Nagarajan [1] find that the per-parameter learned regularization that we use (ARD) is related to a complex form of per-weight L1 regularization. Such a per-weight L1 regularization is, however, impossible to realize in practice as it has too many hyperparameters.
> > >
> > > [1] Wipf, D., & Nagarajan, S. A new view of automatic relevance determination. NeurIPS 2007.

---

> > > > ### Comment · Reviewer_rzPD · 2024-08-13
> > > >
> > > > Dear authors,
> > > >
> > > > Thank you for the additional comments - the discussion on the L1 regularisation is insightful and really interesting. I'll keep this in mind and discuss further with the other reviewers.

---

### Author Rebuttal · Authors · 2024-08-07

Thank you for the constructive review and feedback. We appreciate the opportunity to clarify and elaborate on the decisions made in our work. One shared comment across two reviews was the choice of the Laplace approximation and the use of the Gaussian distribution.

**Choice of Laplace Approximation and Gaussian Distributions**
The primary reason for utilizing the Laplace approximation and structured Gaussian priors in our framework stems from the need for computational efficiency and scalability. While the marginal likelihood provides a powerful tool for balancing model complexity and data fit, implementing this in a practical and scalable manner is non-trivial and indeed, the combination of Gaussian distributions with Laplace approximations are the only example in the literature known to us for which this works (Immer et al., 2021). For mean-field variational inference, for example, Blundell et al. (2015) found it to not work.

**Computational Feasibility**
Applying more complex priors such as the horseshoe or Indian buffet process priors, as suggested by some reviewers, introduces significant computational overhead. These priors often require sophisticated inference techniques, such as Markov Chain Monte Carlo (MCMC) or variational inference, which are computationally expensive and challenging to scale to high-dimensional, modern architectures. For instance, methods like those proposed by Ghosh et al. (2018) and Kessler et al. (2021) typically focus on much smaller models where computational resources are not as limiting. In contrast, our work aims to provide a scalable solution applicable to large-scale networks commonly used in practice without the need for extensive architectural changes, ensuring support for real-world use cases. This scalability is made possible thanks to the Laplace approximation (Daxberger et al., 2021), which approximates the model's posterior with a Gaussian distribution in a computationally efficient manner, leveraging second-order information of the loss landscape. As we show empirically, this approximation is good enough to yield tangible benefits.

**References**

Immer, A., Bauer, M., Fortuin, V., Rätsch, G., & Khan, M.E. (2021). Scalable Marginal Likelihood Estimation for Model Selection in Deep Learning. ICML.

Blundell, C., Cornebise, J., Kavukcuoglu, K., & Wierstra, D. (2015). Weight uncertainty in neural network. ICML.

Ghosh, S., Yao, J., & Doshi-Velez, F. (2018). Structured Variational Learning of Bayesian Neural Networks with Horseshoe Priors. ICML.

Kessler, D., Aicher, C., & Fox, E. B. (2021). Indian Buffet Process Priors for Bayesian Neural Networks. NeurIPS.

Daxberger, E., Kristiadi, A., Immer, A., Eschenhagen, R., Bauer, M., & Hennig, P. (2021). Laplace Redux – Effortless Bayesian Deep Learning. NeurIPS.

**Adressing specific review**
To address all comments, one review in particular seems to misunderstand a few key aspects of our work, thus we would like to clarify them further.
> Novelty

We show that automatic relevance determination, which leads to a prunable neural network, is possible for deep learning models for the first time. Our method is scalable and leverages Bayesian properties to enhance the prunability of neural networks and generalizes to different architectures. Our approach includes a new provably optimal approximation for diagonal priors in the KFAC eigenbasis, confirming that diagonal priors improve prunability. We introduce a one-shot pruning approach that eliminates the need for parameter-linked pruning hyperparameters but automatically learns them, and OPD, a cost-effective pruning criterion akin to the popular OBD. OPD performs on par or better than many other criteria in practice. This method can also be applied to pre-trained networks, as demonstrated for GPT-2 on IMDB (detailed in appendix B.7.3).
> The baseline methods used are too simplistic. Baselines such as IMP, RigL, or DST should be included.

The criteria used are not baselines, as our claim is that SpaM works with any pruning criterion to achieve high sparsifiability. The only real baseline is MAP, compared directly with SpaM. We argue that SpaM could be combined with all modern pruning criteria, which is beyond our work's scope.
Our primary focus is to present a practical sparsification pipeline that aligns with various architectures with minimal hyperparameters. Methods like IMP, involving iterative retraining and weight resetting, can be computationally expensive and sensitive to initialization [1]. Even with multiple iterations of train, prune, and retrain, our results outperform theirs, as seen in Figure 14 for ResNet-18 on CIFAR-10.

DST and RigL require hyperparameter tuning for each architecture, adding to the computational cost and being impractical for pre-trained networks. RigL requires tuning sparsity level, update interval, growth method, pruning ratio, and initial sparsity, while DST involves similar hyperparameters and added growth allocation and redistribution methods(some criteria used: magnitude, gradient contribution).

In contrast, SpaM uses a fully automatic one-shot pruning approach, avoiding costly retraining. Our experiments include well-established criteria like magnitude and random pruning, and strong criteria like SNIP, SynFlow, and GraSP. Notably, GraSP, as in our work, has shown high pruning performance on various benchmarks [2,3]. These comparisons showcase our method's effectiveness in a practical setting, demonstrating that SpaM-trained networks perform better than those trained using MAP at extreme sparsities.

[1] Frankle & Carbin (2019). "The Lottery Ticket Hypothesis: Finding Sparse, Trainable Neural Networks." International Conference on Learning Representations (ICLR).

[2] Rachwan et al. (2022) . "Winning the Lottery Ahead of Time: Efficient Early Network Pruning." ICML.

[3] Lubana & Dick (2021). "A Gradient Flow Framework for Analyzing Network Pruning." International Conference on Learning Representations (ICLR).

---

### Decision · Program_Chairs · 2024-09-25

**Decision:**

Accept (poster)

**Comment:**

This paper introduces a framework called Sparsifiability via the Marginal likelihood (SpaM) that leverages Bayesian marginal likelihood with sparsity-inducing priors to enhance the sparsification of neural networks. All reviewers praised the originality and effectiveness of Bayesian marginal likelihood as a pruning criterion, and most reviewers also appreciated the thoroughness of the experiments performed. Some reviewers questioned the choice of Laplace approximation and Gaussian approximation and asked for comparison with previous works featuring other sparsity-inducing priors like horseshoe priors; the authors mainly defended from the perspective of computational feasibility and the fact that some of the prior works require more invasive changes to the architectures/training protocols, which largely addressed the reviewers' concerns. On this point, while the AC also does not see this as a major issue post-rebuttal and acknowledges the computational concerns, the AC still thinks that some empirical comparison against these prior works, even on smaller scale setups where these baselines are indeed applicable, could still be helpful to strengthen the arguments in the paper.

Other concerns were raised on novelty (although the AC does not view this as a major weakness given that all reviewers acknowledged the originality and effectiveness) and comparison against approaches with type-II MLE, but the authors have answered the questions satisfactorily.

Post-rebuttal, all reviewers unanimously recommend acceptance and the AC concurs with this recommendation. The authors are encouraged to incorporate all requested changes and discussions into the camera-ready version of the paper.